# Suppression of magnetic ordering in XXZ-type antiferromagnetic monolayer NiPS$_3$

Kangwon Kim[1], Soo Yeon Lim[1], Jae-Ung Lee [1], Sungmin Lee[2,3], Tae Yun Kim[3,4], Kisoo Park[2,3], Gun Sang Jeon[5], Cheol-Hwan Park [3,4], Je-Geun Park [2,3] & Hyeonsik Cheong [1]

How a certain ground state of complex physical systems emerges, especially in two-dimensional materials, is a fundamental question in condensed-matter physics. A particularly interesting case is systems belonging to the class of XY Hamiltonian where the magnetic order parameter of conventional nature is unstable in two-dimensional materials leading to a Berezinskii—Kosterlitz—Thouless transition. Here, we report how the XXZ-type anti-ferromagnetic order of a magnetic van der Waals material, NiPS$_3$, behaves upon reducing the thickness and ultimately becomes unstable in the monolayer limit. Our experimental data are consistent with the findings based on renormalization-group theory that at low temperatures a two-dimensional XXZ system behaves like a two-dimensional XY one, which cannot have a long-range order at finite temperatures. This work provides the experimental examination of the XY magnetism in the atomically thin limit and opens opportunities of exploiting these fundamental theorems of magnetism using magnetic van der Waals materials.

[1] Department of Physics, Sogang University, Seoul 04107, Korea. [2] Center for Correlated Electron Systems, Institute for Basic Science, Seoul 08826, Korea. [3] Department of Physics and Astronomy, Seoul National University, Seoul 08826, Korea. [4] Center for Theoretical Physics, Seoul National University, Seoul 08826, Korea. [5] Department of Physics, Ewha Womans University, Seoul 03760, Korea. Correspondence and requests for materials should be addressed to C.-H.P. (email: cheolhwan@snu.ac.kr) or to J.-G.P. (email: jgpark10@snu.ac.kr) or to H.C. (email: hcheong@sogang.ac.kr)

I t is an enduring theme of physical science how a certain ground state emerges out of often complex underlying principles of nature. Our understanding of this fundamental question captures all the essence of what we know about the system, be it cosmos or real materials. One prime example is magnetism in two-dimensional (2D) systems. Unlike in one- or three-dimension, fluctuations are sensitive to the symmetry of the order parameters: Ising, XY, and Heisenberg types. Of the three types, the Ising Hamiltonian was the first to be solved by Onsager[1]. About 20 years later, there were theoretical breakthroughs for the Heisenberg model[2–4].

For the XY model, the trio of Berezinskii[5] and Kosterlitz and Thouless[6] demonstrated independently that the 2D XY system hosts a very unusual ground state of an algebraic order at low temperatures through what is now known as the Berezinskii−Kosterlitz−Thouless (BKT) transition. The generic form of the magnetic Hamiltonian can be written as follows[7]:

$$H = -\sum_{\langle i,j \rangle} \left( J_x S_i^x S_j^x + J_y S_i^y S_j^y + J_z S_i^z S_j^z \right), \quad (1)$$

where $J_{x,y,z}$ is the nearest-neighbor exchange interaction for spin components, and $i$ and $j$ run through all lattice sites and nearest neighbors, respectively. $S_i^x$, $S_i^y$, and $S_i^z$ are the $x$, $y$, and $z$ component of the total spin at $i$-site. The critical behaviors of 2D magnetic systems have been studied using layered magnetic crystals or ultra-thin metal films[8]. However, it is still desirable for one to investigate the major features of 2D magnetism using true 2D material. For the 2D Ising system ($J_x = J_y = 0$) the magnetic ground state is stable even in the 2D limit, whose experimental evidence has been recently presented using magnetic van der Waals materials: $FePS_3$ [9,10] with antiferromagnetic order and $Cr_2Ge_2Te_6$ [11] and $CrI_3$ [12] with ferromagnetic order.

With the success in the recent experimental investigation of 2D Ising systems, one can think of using these newly found magnetic van der Waal materials for the studies of the XY model ($J_x = J_y, J_z = 0$), which is far more interesting and expected to host much richer physics. One can generalize the problems for two dimensions by using the XXZ model ($J_x = J_y \neq J_z$). We note that from renormalization-group studies[13,14], an antiferromagnetic 2D quantum-spin system with easy-plane-like anisotropy, no matter how small the anisotropy is, behaves like an XY system at low temperatures. With the advent of a new class of magnetic van der Waals materials, one at last seems to have the right materials to start with[15,16].

Transition metal phosphorus trisulfides (TMPS$_3$, TM = V, Mn, Fe, Co, Ni, or Zn) are a new class of antiferromagnetic van der Waals materials that are suitable for studying antiferromagnetic ordering in the 2D limit: $FePS_3$ of Ising-type, $NiPS_3$ of XY or XXZ-type, and $MnPS_3$ of Heisenberg-type[17,18]. They can be easily exfoliated into few atomic layers[19], allowing one to examine the dependence of the magnetic ordering on the dimensionality. There still remains one experimental difficulty: since antiferromagnets do not have a finite net magnetization, detection of antiferromagnetic ordering in few-layer samples, let alone monolayer, is extremely challenging[15].

Raman spectroscopy has proven to be a powerful technique for the studies of 2D materials. Since the Raman spectrum changes sensitively with the thickness[20,21], it is suitable not only to determine structural parameters such as the number of layers but also to study thickness-dependent physical properties. In particular, it is possible to study magnetic properties by exploiting the spin dependence of Raman processes. For example, two-magnon Raman scattering is often used to study antiferromagnetism[22,23], and Raman spectroscopy has been successfully used to investigate the Ising-type antiferromagnet FePS$_3$[9,10]. By monitoring the

appearance of a series of new Raman modes due to doubling of the unit cell upon antiferromagnetic ordering, it was found that the Ising-type magnetic order of FePS$_3$ is preserved down to the monolayer limit. In this work, we measured the Raman signatures of the antiferromagnetic ordering in NiPS$_3$ as a function of the number of layers downs to monolayer, and found that the ordering persists down to 2 layers (2L) but is dramatically suppressed in the monolayer (1L) limit. At the same time, there is persistent spin fluctuations even in monolayer samples. These experimental findings from monolayer systems are consistent with the theoretical predictions of the XY model[5,6].

## Results

**Crystal and magnetic structures of NiPS$_3$.** Bulk NiPS$_3$ has a monoclinic structure with the point group $C_{2h}$, whereas monolayer NiPS$_3$ has a hexagonal structure with the point group $D_{3d}$[19,24,25]. As shown in Fig. 1a, Ni atoms are arranged in a hexagonal lattice, each of them being surrounded by six S atoms with trigonal symmetry. These S atoms are connected to two P atoms located above and below the Ni plane. Two P atoms and six S atoms are covalently bonded among themselves, forming a $(P_2S_6)^{4-}$ anion complex of a pyramidal structure. The layers are weakly bound to each other by van der Waals interaction along the $c$-axis and can be easily exfoliated to atomically thin few-layer samples[19]. Figure 1b is an atomic force microscope image of one of the exfoliated monolayer NiPS$_3$ samples measured in this work (see Supplementary Fig. 1 for samples with other thicknesses).

Recent neutron diffraction studies have shown that below Néel temperature ($T_N$) spins in bulk NiPS$_3$ are aligned mostly in the $ab$ plane with a small component along the $c$-axis[17,18], which is consistent with the temperature dependence of the magnetic susceptibility (see Supplementary Fig. 2). The ordered magnetic moments appear to point more towards the $a$-axis than the $b$-axis and the same-spin chains are aligned along the zigzag direction[17,18] as shown by the red arrows in Fig. 1a. We note that due to the $ab$ anisotropy, this is not an exact XXZ system, but an approximate one.

**Raman signatures of the antiferromagnetic phase transition.** Figure 2 compares polarized Raman spectra of bulk NiPS$_3$ in two different phases: antiferromagnetic phase at $T = 10$ K and paramagnetic phase at $T = 295$ K. All the peaks in the Raman spectra are labeled as P$_1$, P$_2$, etc. in the order of increasing frequency. As bulk paramagnetic NiPS$_3$ belongs to the $C_{2h}$ point group[25], the zone center phonon modes are represented by $\Gamma = 8A_g + 7B_g + 6A_u + 9B_u$. In the backscattering geometry, both the $A_g$ and $B_g$ modes are Raman allowed in the parallel-polarization scattering configuration $[\bar{z}(xx)z]$ whereas only the $B_g$ modes are active in the cross-polarization scattering configuration $[\bar{z}(xy)z]$. The Raman modes at higher frequencies are mostly attributed to the intramolecular vibrations from $(P_2S_6)^{4-}$ bipyramid structures: similar features were previously observed for FePS$_3$ [9]. On the other hand, the low-frequency peaks (P$_1$ and P$_2$) are due to vibrations involving the heavy Ni atoms[25].

The most prominent difference between paramagnetic ($T = 295$ K) and antiferromagnetic ($T = 10$ K) phases is the appearance of the broad peak centered at ~550 cm$^{-1}$ in the antiferromagnetic phase due to two-magnon scattering, or the double spin-flip processes via the exchange mechanism in antiferromagnets with the collinear structure[22,26]. The spin orientation of Ni atoms lies largely on the $ab$ plane with some $c$-axis component due to the magnetic ordering at low temperature, and the two-magnon signals become well defined. We note that the two-magnon Raman signals are consistent with our calculations of the two-magnon spectrum using the XXZ Hamiltonian (see

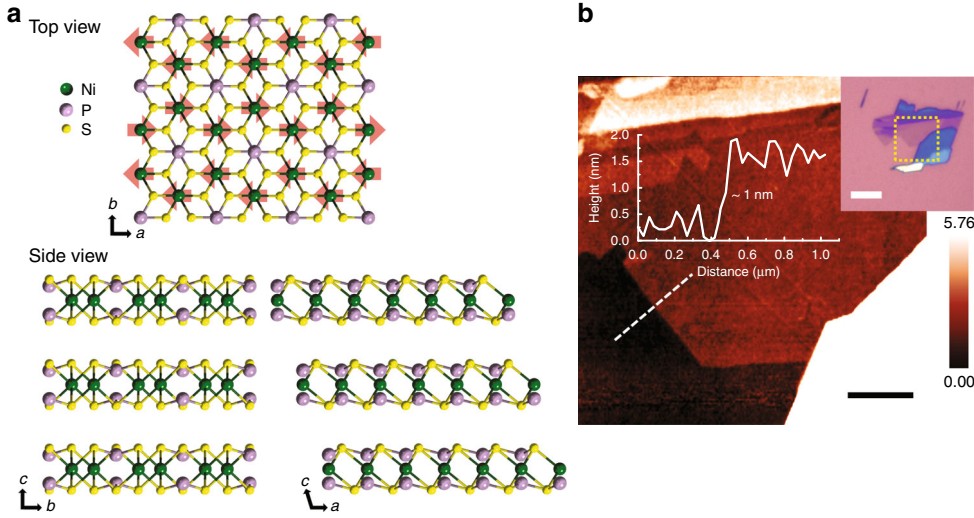

**Fig. 1** Magnetic van der Waals material NiPS$_3$. **a** Crystal structure of NiPS$_3$. Red arrows indicate the spin orientations of Ni atoms below Néel temperature ($T_N$). **b** Atomic force microscope image and thickness line profile of monolayer NiPS$_3$. Optical image of the sample is shown in inset. The black and white scale bars are 1 and 5 μm, respectively

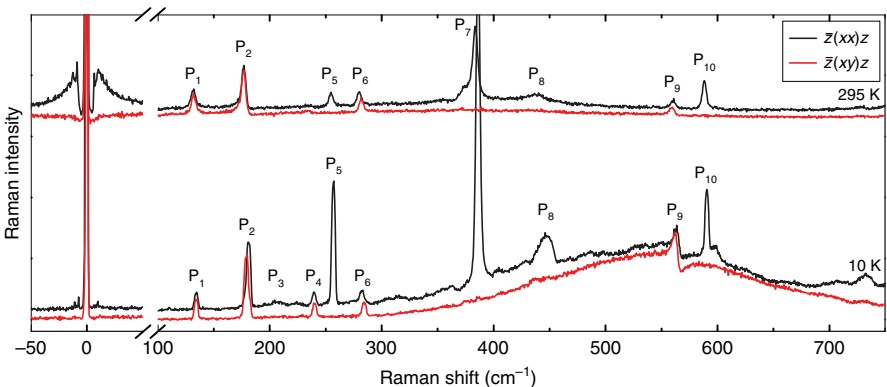

**Fig. 2** Raman spectra of bulk NiPS$_3$. Comparison of Raman spectra measured at $T = 10$ and 295 K in parallel [$\bar{z}(xx)z$] (black) and cross [$\bar{z}(xy)z$] (red) polarization scattering configurations

Supplementary Note 1 and Supplementary Fig. 3). Also, the clear signals centered at 0 cm$^{-1}$ in the spectrum obtained in the parallel-polarization configuration at $T = 295$ K are ascribed to quasi-elastic scattering (QES) from magnetic fluctuations, which is often observed in low-dimensional spin systems[23,27,28]. These QES signals become considerably weakened at lower temperatures since the spin fluctuations are suppressed in the magnetically ordered phase. Unlike the two-magnon signal which does not depend on the polarization, the QES signal is much stronger in parallel polarization, presumably because low-energy spin fluctuations do not change the polarization of the scattered photon very much. In addition, the peak P$_9$ shows a prominent Breit−Wigner−Fano (BWF) line shape due to Fano resonance at low temperatures[29]. Fano resonance requires quantum interference between a discrete excitation and a continuum[30] and the BWF line shape is described by refs. [30,31]:

$$I(\omega) = I_0 \frac{[1 + 2(\omega - \omega_0)/(q\Gamma)]^2}{[1 + 4(\omega - \omega_0)^2/\Gamma^2]}, \quad (2)$$

where $\omega_0$ is the bare phonon frequency, $\Gamma$ the linewidth, and $q$ the asymmetry parameter, where $|1/q|$ correlates with the strength of the coupling. In this case, the discrete excitation is the emission of a phonon, whereas the continuum corresponds to the broad two-

magnon excitation. We note that P$_9$ is the only peak that appears in the Raman spectra obtained in the cross-polarization configuration among the peaks that have significant overlap with the two-magnon continuum.

According to our calculations (See Supplementary Table 1 and Supplementary Fig. 4), the phonon modes associated with the P$_9$ peak are two almost-degenerate $E_g$-like modes. Thanks to the in-plane $E_g$-like nature of these modes, they appear in both parallel- and cross-polarization Raman scattering configurations. Our Raman data show that in the energy range where the two-magnon continuum is strong, only these $E_g$-like modes couple strongly to the two-magnon continuum at low temperatures. This finding is indeed in good agreement with Rosenblum et al.[29]. Similar strong phonon−magnon coupling for in-plane phonons was also observed in (Y,Lu)MnO$_3$, a model compound for spins on a 2D triangular lattice[32]. We note that although P$_5$ and P$_7$ show strong relative enhancement at low temperatures, their peak positions or intensities do not show any correlation with magnetic ordering (see Supplementary Fig. 5).

**Two-magnon scattering and Fano resonance of NiPS$_3$.** In order to study the thickness dependence of the antiferromagnetic ordering, we measured the temperature dependence of the Raman spectra as a function of the number of layers down to the

monolayer limit. Figure 3a shows the temperature dependence of the Raman spectrum of bulk $NiPS_3$ measured in the cross-polarization configuration. Suppression of several phonon peaks makes it easier to monitor the two-magnon signals in the cross-polarization configuration. In these data, we observe that the two-magnon signals gradually grow and shift towards higher frequencies as the temperature decreases below $T_N$. These observations are typical of antiferromagnetic materials, where the two-magnon signals redshift and become broader and weaker as the spectral weight is shifted to QES when the temperature is increased[27,28,33,34]. We have carried out similar measurements for few-layer samples with different thicknesses (see Supplementary Fig. 6) and the results are summarized in Fig. 3c–f. The peak position, intensity, and width of the two-magnon signals show little dependence on the number of layers from bulk all the way down to two layers. The coupling strength of Fano resonance, represented by $|1/q|$, also shows dramatic enhancement below bulk $T_N$. On the other hand, the spectrum of monolayer $NiPS_3$ seems to have a qualitatively different temperature dependence from that of samples with other thicknesses. For example, Fig. 3b shows the temperature dependence of the Raman spectrum of monolayer $NiPS_3$ as a function of temperature. Unlike other samples thicker than monolayer, the two-magnon feature is not well defined even when the temperature is much lower than the bulk $T_N$. The position of the two-magnon peak is also considerably lower in frequency than in the case of thicker samples. Furthermore, the peak $P_9$ does not show any indication of Fano resonance. All these observations point to the conclusion that antiferromagnetic ordering is not fully developed in monolayer $NiPS_3$ even at the lowest temperature measured (25 K).

**Magnetic-order-induced frequency difference for $P_2$ phonons**. Interestingly enough, a closer inspection of the bulk spectra in Fig. 2 reveals that $P_2$ comprises two peaks at low temperatures as the peak position is slightly different under different polarization configurations. The higher frequency peak $\left(P_2^{\parallel}\right)$ appears in the parallel-polarization configuration, while the lower frequency peak $\left(P_2^{\perp}\right)$ is seen in the cross-polarization configuration (see Supplementary Fig. 7 for complete polarization dependence data). We should emphasize that there is minimal difference in the phonon frequency in the data taken at $T = 295$ K. Figure 4a shows the temperature dependence of the polarized Raman spectra of bulk $NiPS_3$ in the range 120–300 $cm^{-1}$. We can see a clear sign of the temperature-dependent phonon frequency difference for $P_2$. On the other hand, $P_6$ also shows a small peak frequency difference in two polarization configurations, which does not seem to depend much on the temperature (see Fig. 4b for the summary). If we compare the phonon frequency difference of $P_2$ with the susceptibility data (Fig. 4c), the cross correlation between them is evident.

In order to elucidate the origin of this phonon frequency difference, theoretical calculations of the vibrational modes in the ordered phase were carried out using density functional theory (DFT) with the frozen-phonon method. It was found that near the frequency of $P_2$, there are two phonon modes with similar frequencies but with different symmetries. These modes originate from the doubly degenerate $E$-like modes of monolayer $NiPS_3$, which are split due to monoclinic stacking of the layers. However, due to the weak interlayer coupling, this splitting is expected to be very small in the paramagnetic phase[25]. Experimentally, we observed almost zero separation between the two peaks measured in the parallel- and cross-polarization configurations in the paramagnetic phase. We further note that we did not observe any superlattice peaks from our single crystal X-ray diffraction (XRD) experiments performed at temperatures as low as 20 K. Therefore,

we can conclude that the temperature dependence of the splitting of $P_2$ arising from structural symmetry breaking should be very small, if at all. Figure 4d illustrates these two vibrational modes in the antiferromagnetic phase. The $A_g$ mode involves vibrations of the Ni atoms in the periodic direction of the same-spin zigzag chains (see Fig. 1a), whereas the $B_g$ mode involves Ni atoms vibrating parallel to this direction. Owing to the relative directions of the vibration and the distribution of the spin polarization, the vibrational frequencies shift differently upon antiferromagnetic ordering and results in observed increase in the phonon frequency difference $\Delta P_2$, which is mostly of magnetic origin. Since the other peaks do not show any dramatic changes with temperature, one can also exclude a structural phase transition as the origin of this phonon frequency difference. We note that a similar phonon frequency difference below critical temperature[35] has been reported for $Cr_2Ge_2Te_6$.

We note that in the spectra for thinner samples there are several peaks that originate from multi-phonon scattering at ~210, ~590, and ~800 $cm^{-1}$ (see Supplementary Fig. 8). Some of these peaks are strongly enhanced due to the resonance effect. Similar enhancement effects on multiphonon peaks due to resonance have been observed in other 2D materials[36–38]. For example, the relative intensities or the line shapes of $P_3$ at ~210 $cm^{-1}$ or the peak at ~590 $cm^{-1}$ vary with the excitation laser energy, which is a clear indication of resonant processes (see Supplementary Note 2 and Supplementary Fig. 9). We further checked the temperature dependence of $P_3$ for 1−4L $NiPS_3$ (see Supplementary Fig. 10). Since the frequency and the intensity of $P_3$ does not exhibit any significant change near the Néel temperature, we can safely conclude that $P_3$ is not related to the magnetic ordering.

**Estimation of Néel temperature of few-layer $NiPS_3$ from $\Delta P_2$**. Now, we would like to inspect this frequency difference as a function of temperature for few-layer $NiPS_3$ (see Supplementary Fig. 11). Figure 5a shows temperature dependence of $\Delta P_2$ for various thicknesses including monolayer. The antiferromagnetic transition temperatures are extracted by using the spin-induced phonon frequency shift model[39,40] (see Supplementary Note 3 and Supplementary Fig. 12 for details) and summarized in Fig. 5b. As seen in Fig. 5a, $\Delta P_2$ shows a clear onset for two-layer or thicker samples and the transition temperature gets slightly lower as the thickness decreases. On the contrary, $\Delta P_2$ for monolayer $NiPS_3$, if any, is virtually temperature independent. These results confirm our conclusion from the analysis of the two-magnon signals: (1) the antiferromagnetic transition temperature ($T_N$) depends only slightly on the thickness for two-layer or thicker $NiPS_3$, and (2) antiferromagnetic ordering is significantly suppressed in monolayer $NiPS_3$.

In order to rule out the possible extrinsic effect due to the substrate, we fabricated additional samples on hexagonal boron nitride (hBN) layers and compared them with those samples prepared on $SiO_2/Si$ substrates. We found that there is virtually no difference between the results for samples exfoliated directly on $SiO_2/Si$ and those for samples on $hBN/SiO_2/Si$ (see Supplementary Figs. 13 and 14), which strongly indicate that the extrinsic effect due to the substrate is not important.

**Thickness dependence of QES signals**. We further analyze the low-frequency QES signals due to the magnetic fluctuations[23,26]. Since such QES signals are suppressed in the presence of the spin ordering in ferromagnetic or antiferromagnetic materials, its suppression can also be used as another good indicator for magnetic ordering. Figure 6a, b compares the temperature dependence of the low-frequency region of the polarized Raman

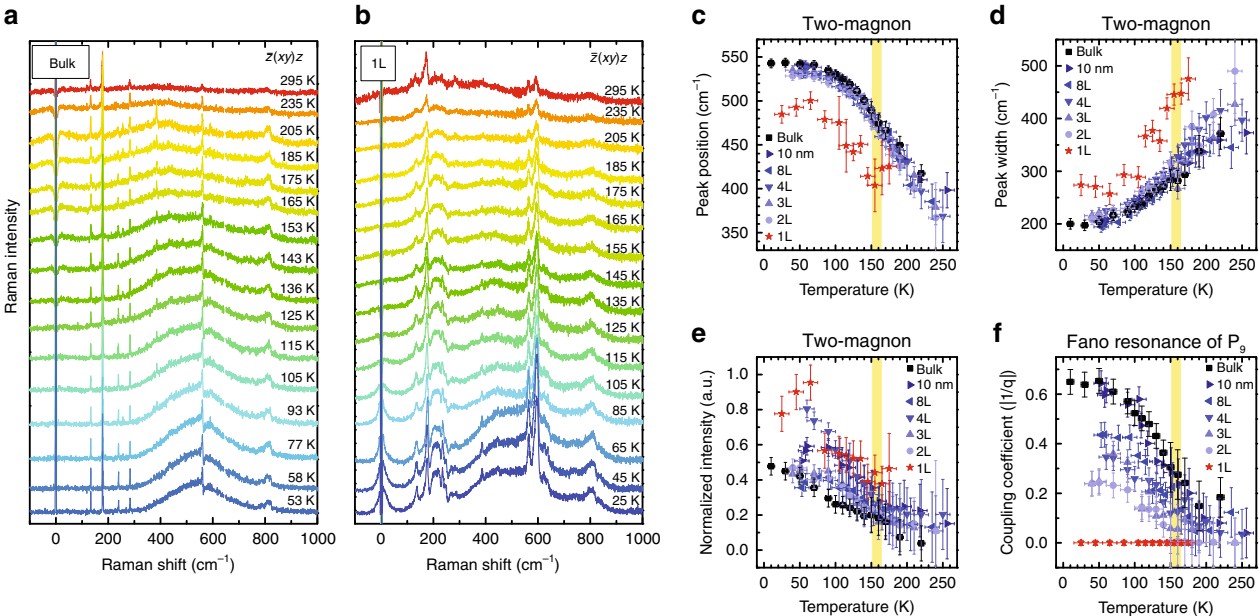

**Fig. 3** Temperature dependence of two-magnon signals and Fano resonance of $P_9$. **a, b** Raman spectra of bulk (**a**) and monolayer (**b**) NiPS$_3$ in cross-polarization as a function of temperature. **c–f** Peak position (**c**), width (**d**), and normalized intensity (**e**) of two-magnon signals; and coupling coefficient ($|1/q|$) of Fano resonance of $P_9$ (**f**) as a function of temperature for various thicknesses. Error bars indicate the experimental uncertainties in temperature and the uncertainties in the fitting procedure to determine each parameter

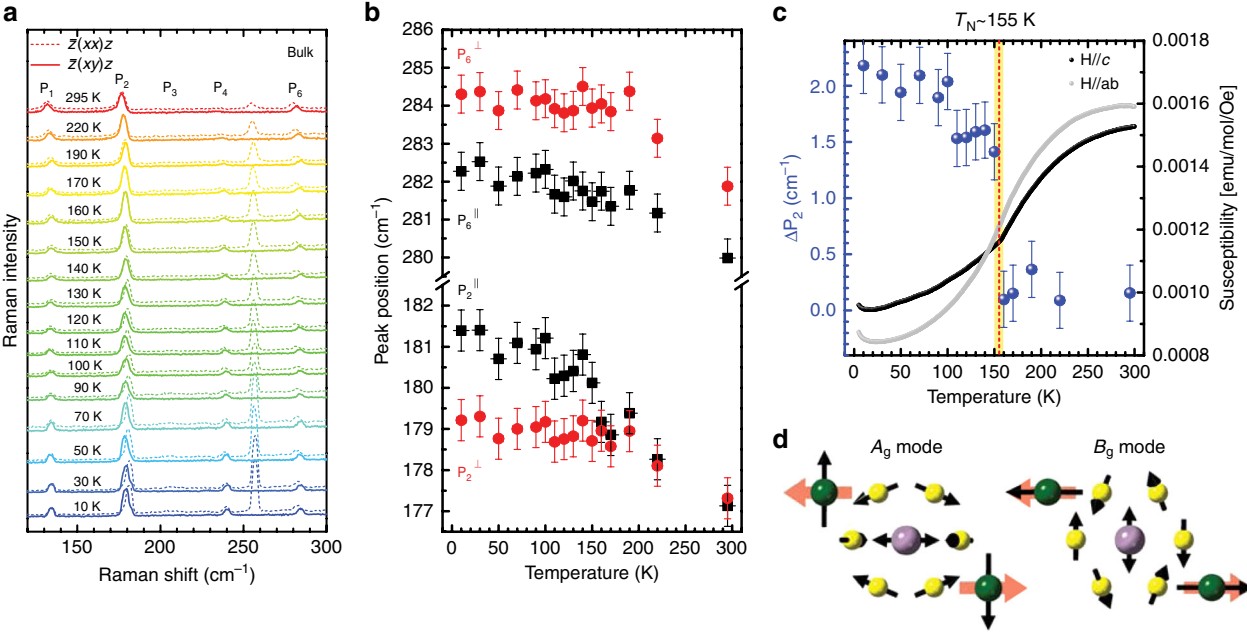

**Fig. 4** Magnetic-order-induced frequency difference for phonon $P_2$. **a** Polarized Raman spectra of bulk NiPS$_3$ as a function of the temperature. **b** Peak positions of $P_2$ and $P_6$ obtained in parallel (black squares, $P_2^{||}$ and $P_6^{||}$) and cross (red circles, $P_2^{\perp}$ and $P_6^{\perp}$) polarization configurations as a function of the temperature. **c** Temperature dependences of phonon frequency difference $\Delta P_2$ ($\Delta P_2 \equiv \left| P_2^{||} - P_2^{\perp} \right|$, blue circles) and susceptibility of bulk NiPS$_3$. The error bars indicate experimental uncertainties. **d** Schematics of lattice-vibration patterns associated with $A_g$ and $B_g$ modes near ~180 cm$^{-1}$. Black arrows indicate the direction of atomic displacement and thick red arrows indicate spin orientations of Ni atoms below $T_N$

spectra of 9L and monolayer NiPS$_3$. In parallel-polarization configuration, the QES signals from 9L NiPS$_3$ are strongly suppressed below $T_N$, but a considerable amount of spectral weight persists down to the lowest temperature in the case of monolayer NiPS$_3$. The difference is even more striking for the case of cross-polarization configuration. Here, the QES signals from 9L NiPS$_3$ are somewhat enhanced near $T_N$, presumably because of strong

spin fluctuations near the phase transition[27]. On the other hand, the QES signals from monolayer NiPS$_3$ monotonically increase as the temperature is lowered as if it is approaching the phase transition. The temperature dependences of QES signals for intermediate thicknesses, such as 2L and 3L NiPS$_3$, are qualitatively similar to that of 9L NiPS$_3$ (see Supplementary Fig. 15). For a more quantitative analysis, the temperature dependence of the

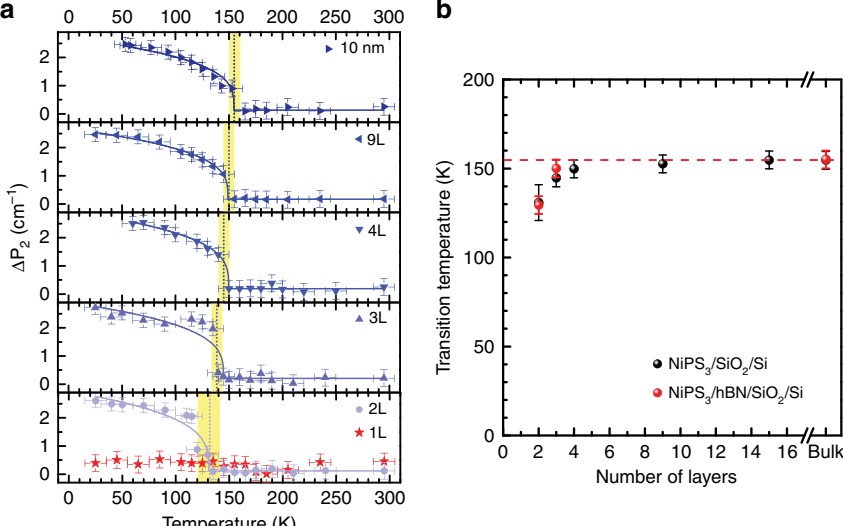

**Fig. 5** Thickness dependence of Néel temperature for few-layer NiPS₃. **a** Temperature dependence of magnetic-order-induced frequency difference $\Delta P_2$ for various thicknesses. The error bars indicate experimental uncertainties, the dashed vertical lines indicate the Néel temperature for each thickness, and the solid curves are fitting results by using the spin-induced phonon frequency shift model. **b** Estimated antiferromagnetic transition temperature for various thicknesses by using $\Delta P_2$ of NiPS₃ samples on SiO₂/Si (black) and hBN/SiO₂/Si (red) substrates. The error bars indicate experimental uncertainties, and the red dashed line shows bulk Néel temperature (155 K)

phonon population should be considered. Although the exact mechanism of QES in low-dimensional systems is not fully understood yet[23], we follow the theory of Reiter[41] and Halley[42] to analyze our data. The measured intensity of Stokes-shifted QES is simply expressed by[40–45]

$$I(\omega) \propto \frac{\omega}{1 - e^{-\hbar\omega/k_{\mathrm{B}}T}} \frac{C_{\mathrm{m}} T \, D k^2}{\omega^2 + (Dk^2)^2}, \qquad (3)$$

where $C_{\mathrm{m}}$ is the magnetic specific heat and $D$ is the thermal diffusion constant $D = K/C_{\mathrm{m}}$ with the magnetic contribution to the thermal conductivity $K$. Since the Raman response $\chi''(\omega)$ for Stokes scattering is given by $\chi''(\omega) = I(\omega)/[n(\omega) + 1]$, where $n(\omega) = [\exp(\hbar\omega/k_{\mathrm{B}}T) - 1]^{-1}$ is the Bose−Einstein factor, Eq. (3) is expressed in terms of the Raman response $\chi''(\omega)$[40,45],

$$\frac{\chi''(\omega)}{\omega} \propto C_{\mathrm{m}} T \frac{D k^2}{\omega^2 + (Dk^2)^2}. \qquad (4)$$

We obtained the spectral weight of QES by integrating the normalized Raman response $\chi''/\omega$ in the range of 11–40 cm⁻¹. Figure 6c shows the temperature dependence of the spectral weight of QES in both parallel- and cross-polarization configurations for several thicknesses. For both polarization configurations, the spectral weight of QES shows a peak near the magnetic transition temperature for all thicknesses except monolayer. In the case of monolayer NiPS₃, the spectral weight of QES almost monotonically increases as the temperature is lowered in both polarization configurations as if it is approaching the phase transition from above. These results are consistent with our prior observation from the raw Raman data in Fig. 6a, b and support our interpretation that antiferromagnetic ordering is drastically suppressed in monolayer NiPS₃. The increase in the spectral weight of QES at low temperatures indicates that spin fluctuations in monolayer NiPS₃, most probably related to the bound vortex−antivortex pairs[5,6], are growing persistently as the temperature is decreasing even down to zero Kelvin.

**Monte Carlo simulations**. In order to shed further light on the experimental observations, we have carried out Monte Carlo simulations as a function of the number of layers (see Supplementary Note 4 and Supplementary Fig. 16). The simulation results qualitatively support our interpretation of the experimental data. The simulated Néel temperature slowly decreases as the thickness is reduced, down to 2L case, but the decrease is more dramatic for monolayer. For a better quantitative agreement, more sophisticated calculations will be necessary, which is beyond the scope of this work.

**Discussion**

NiPS₃, an antiferromagnetic van der Waals material, is a good model system for the XY Hamiltonian. Using this material, we investigated the magnetic signals in the form of two-magnon and QES as a function of the sample thickness down to the monolayer limit. All our experimental observations coherently point to the conclusion that the antiferromagnetic ordering persists down to two-layer samples and is drastically suppressed in the monolayer. Furthermore, the Néel temperature is only slightly dependent on the number of layers as long as it is two or larger. This result is in stark contrast to the case of Ising-type antiferromagnet FePS₃, in which antiferromagnetic ordering persists down to the monolayer limit[9]. It is also different from the case of recently reported ferromagnetic 2D materials such as CrI₃ or Cr₂Ge₂Te₆, of which the Curie temperature decreases as the number of layers decreases but remains finite in the monolayer limit[11,12]. The thickness dependence strongly indicates that the intra-layer exchange interactions are much stronger than the interlayer ones. However, the interlayer interaction suppresses the logarithmically divergent spin fluctuations, except in the monolayer. In monolayer NiPS₃, where the static and bulk antiferromagnetic order is suppressed by the strong fluctuations, there is the low-temperature enhancement of QES. We note that all these experimental observations are in good agreement with the theoretical predictions of the XY model. Our work, one of the very rare experimental realizations of the XY Hamiltonian in the atomically thin monolayer limit, opens a new window of opportunities to study

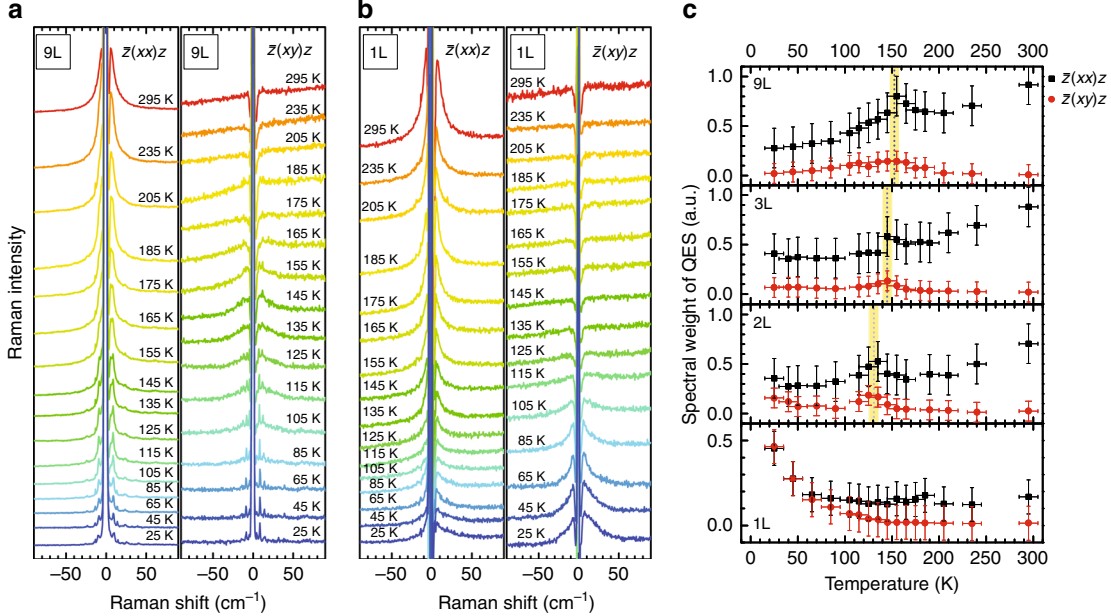

**Fig. 6** Temperature dependence of quasi-elastic scattering signals. **a**, **b** Low-frequency polarized Raman spectra of 9L (**a**) and monolayer (**b**) NiPS$_3$. **c** Spectral weight of QES between 11 and 40 cm$^{-1}$ as a function of temperature for various thicknesses for parallel (black squares) and cross (red circles) polarization scattering configurations. The error bars indicate experimental uncertainties and dashed lines show estimated transition temperatures by using $\Delta P_2$

the extremely rich physics of the XY Hamiltonian and probably the BKT transition using a real material.

## Methods

**Sample preparation**. NiPS$_3$ crystals were grown by chemical vapor transport reaction. Inside an argon-filled glove box, elemental powders (purchased from Sigma-Aldrich) of nickel (99.99% purity), phosphorus (99.99%), and sulfur (99.998%) were weighed and mixed in stoichiometric ratio 1:1:3 and an additional 5 wt% of sulfur. The samples were immediately subjected to chemical analysis using a COXI EM-30 scanning electron microscope equipped with a Bruker QUANTAX 70 energy dispersive X-ray system to find the good stoichiometry of all the samples used in this study. The phase purity was further checked by taking powder XRD patterns with Bruker D8 Discover as well as single crystal XRD (XtaLAB P200, Rigaku) as shown in Supplementary Fig. 17. We further characterized the magnetic properties using a commercial setup (the SQUID magnetometer, Quantum Design) as shown in Supplementary Fig. 2.

Few-layer NiPS$_3$ samples were prepared on Si substrates with a layer of 285-nm SiO$_2$ by mechanical exfoliation from a bulk single-crystal NiPS$_3$. Since few-layer NiPS$_3$ samples can be degraded in ambient conditions, the samples were kept in a vacuum desiccator to prevent possible degradations after exfoliation. All the measurements were carried out with the sample in vacuum in order to avoid any degradation during measurements. Atomic force microscopy images taken before and after Raman measurements confirmed that degradation is minimal as long as the sample is kept in vacuum and the excitation laser intensity is kept at 100 μW or less (see Supplementary Note 5 and Supplementary Fig. 18 for details). The number of layers was determined based on the optical contrast[19], atomic force microscopy, and low-frequency Raman measurements (see Supplementary Fig. 1).

**Linear spin-wave theory**. The calculations of one-magnon spectrum were obtained by using SpinW package[46] for the zigzag magnetic ground state, assuming the following Hamiltonian:

$$
\begin{aligned}
H = J_1 \sum_{\langle i,j \rangle} & \left[ S_i^x S_j^x + S_i^y S_j^y + \alpha S_i^z S_j^z \right] \\
+ J_2 \sum_{\langle\langle i,k \rangle\rangle} & \left[ S_i^x S_k^x + S_i^y S_k^y + \alpha S_i^z S_k^z \right] \\
+ J_3 \sum_{\langle\langle\langle i,l \rangle\rangle\rangle} & \left[ S_i^x S_l^x + S_i^y S_l^y + \alpha S_i^z S_l^z \right], \\
+ \sum_i & \left[ D_1 \left( S_i^x \right)^2 + D_2 \left( S_i^z \right)^2 \right]
\end{aligned}
\tag{5}
$$

where the first three terms denote the XXZ Hamiltonian up to third nearest neighbor. The $(x, y, z)$ coordinate system is defined with the known ordered moment of NiPS$_3$ in Supplementary Fig. 3c. The last terms in the bracket are single-ion anisotropy along the $x$- and $z$-axis, respectively. Using the following set

of parameters: $J_1 = 3.18$ meV, $J_2 = 4.82$ meV, $J_3 = 9.08$ meV, $\alpha = 0.66$, $D_1 = -0.89$ meV, $D_2 = 2.85$ meV, we calculated single magnon dispersion and then two-magnon continuum with the kinematic constraints. See Supplementary Information for further details.

**Raman measurements**. All the Raman measurements were carried out in vacuum using an optical cryostat (Oxford Microstat He2) at temperatures from 10 to 295 K. The 514.4-nm (2.41 eV) line of diode-pumped-solid-state (DPSS) laser was used as the excitation source. The laser power was kept below 100 μW to avoid damaging the samples. The laser beam was focused onto the sample by a ×40 microscope objective lens (0.6 N.A.), and the scattered light was collected and collimated by the same objective. The scattered signal was dispersed by a Jobin-Yvon Horiba iHR550 spectrometer (2400 grooves/mm) and detected by a liquid-nitrogen-cooled back-illuminated charge-coupled-device (CCD) detector. Volume holographic filters (Ondax and Optigrate) were used to clean the laser lines and reject the Rayleigh-scattered light. For polarized Raman measurements, an achromatic half-wave plate was used to rotate the polarization of the incident linearly polarized laser beam. In addition, the analyzer angle was used to selectively pass scattered photons with parallel or cross polarizations. Another achromatic half-wave plate was placed in front of the entrance slit to keep the polarization direction of the signals entering the spectrometer constant with respect to the groove direction of the grating[47]. The Raman spectrum of the substrate (SiO$_2$/Si) was measured from the same location without samples at each temperature and subtracted from the sample spectrum after normalization by the intensity of the 520 cm$^{-1}$ silicon phonon peak. The temperature dependence of the Raman spectrum of a bulk NiPS$_3$ crystal was measured separately in a macro-Raman system by using a closed-cycle He cryostat. The excitation laser was focused by a spherical lens ($f = 75$ mm) to a spot of size ~50 μm with a power of 2 mW.

**Monte Carlo simulations**. We calculated the zigzag-type antiferromagnetic order parameter and magnetic susceptibility in bulk and few-layer stacked honeycomb lattice by Monte Carlo simulations. We treated the spins classically and incorporated the short-range exchange interactions up to third neighbors with the anisotropy parameter $\alpha$ extracted from linear spin-wave theory as well as the interlayer coupling. In Monte Carlo simulations we have performed importance sampling with Metropolis algorithm and used simulated annealing for finite-temperature calculation. Typically, at each temperature the first $10^5$ Monte Carlo steps are discarded for equilibrium and the following $10^6$ Monte Carlo steps are used for averaging physical quantities. For all the numerical data from Monte Carlo simulations the numerical uncertainties are smaller than or comparable to the size of symbols. See Supplementary Information for further details.

**Density functional theory (DFT) phonon calculations**. We calculated the phonon modes of monolayer NiPS$_3$ using DFT and frozen-phonon method. Vacuum layers of 12 Å thick were inserted between two adjacent monolayers of NiPS$_3$ to

avoid spurious interactions between the periodic replicas. The lattice parameters and atomic coordinates were fully relaxed by using the Quantum ESPRESSO package[48]. Ion−electron interactions were simulated by using norm-conserving pseudopotentials[49,50]. The exchange-correlation energy was calculated by using the Perdew−Burke−Ernzerhof functional[51]. The kinetic energy cutoff was set to 80 Ry. The Brillouin-zone integrations were carried out by using the $6 \times 6 \times 1$ Monkhorst-Pack grid[52]. The correlation effects of Ni $3d$ electrons were considered by using the density functional theory + U method[53]. We used 4 eV for the effective Hubbard U of Ni $3d$ electrons. The phonon frequencies and eigenmodes were calculated by using the Phonopy package[54].

## Data availability

The data that support the findings of this study are available from the corresponding authors upon request.

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

## Acknowledgements

The authors thank insightful discussions with H.C. Lee, S. Yoon, K. Burch, A. Wildes, and K.-Y. Choi. We are also grateful to S. Kang for his low-temperature single crystal X-ray diffraction experiments. This work was supported by the National Research Foundation (NRF) grants funded by the Korean government (MSIT) (NRF-2016R1A2B3008363, NRF-2018R1D1A1B07048749, No. 2016R1A1A1A05919979, and No. 2017R1A5A1014862, SRC program: vdWMRC center), by a grant (2013M3A6A5073173) from the Center for

Advanced Soft Electronics under the Global Frontier Research Program of MSIT, and by the Creative-Pioneering Research Program through Seoul National University. Computational resources have been provided by KISTI Supercomputing Center (KSC-2017-S1-0011). Work at IBS CCES was supported by Institute for Basic Science (IBS) in Korea (IBS-R009-G1).

## Author contributions

J.-G.P. and H.C. conceived the experiments. S.L. grew bulk NiPS$_3$ crystals. K.K., S.Y.L. and J.-U.L. carried out Raman measurements. K.P. performed spin-wave calculations. T.Y.K. and C.-H.P. carried out first-principles phonon calculations. G.S.J. conducted Monte Carlo simulations. All authors discussed the data and wrote the manuscript together.

## Additional information

**Competing interests:** The authors declare no competing interests.

**Journal peer review**: *Nature Communications* thanks the anonymous reviewers for their contribution to the peer review of this work. Peer reviewer reports are available.

