## [Peer Review File · Nature Communications]

Reviewers' comments:

Reviewer #1 (Remarks to the Author):

The paper reports the polarized Raman study of the thickness dependent magnetic order in a van der Waals material, NiPS₃, which is believed to be an XXZ-type antiferromagnet. They found that the Raman spectra changed dramatically when NiPS₃ is thinned down to monolayer. Together with the theoretical calculation, the paper claims that the antimagnetic order of NiPS₃ becomes unstable in the monolayer limit. The paper is clearly written, and the scientific results taken at face value would be of sufficient interest to justify publication in Nature communication.

However, in my view the authors have insufficiently accounted for potential artefacts or other possibilities that can affect these delicate measurements. The authors should address the questions I raise below before I can recommend publication of their manuscript:

1. How did the authors confirm the quality of the bulk crystal? How is the $T_n=155$ K extracted?
2. The Raman spectrum is almost the same as bulk down to 2L and suddenly changed at 1L, instead of a gradual evolve, which is somewhat unexpected since 2L is already quite '2D'. Do the authors have any explanation about this? Could the dramatic change of Raman spectrum of 1L NiPS₃ be due to degradation of the flake? The authors should do additional experiments to confirm the stability of 1L NiPS₃ and exclude this possibility.
3. For the Raman spectra of 1L NiPS₃ (Fig. 3b and Fig. S7f), some Raman peaks (~ 200 , ~ 600 , ~ 800 cm⁻¹ in cross-polarization configuration and ~ 210 , ~ 250 cm⁻¹ in the parallel- polarization configuration) still show quite strong temperature dependence. Do the authors know the origin of this Raman modes and their temperature dependences? This seems inconsistent with the conclusion of the suppressed magnetic order at 1L limit.
4. Data quality of 1L NiPS₃ is not that satisfactory, especially for these delicate measurements. Could the authors try to improve signal-to-noise ratio by, for example, increase integration time?

Reviewer #2 (Remarks to the Author):

I have read with great attention the paper by Kangwon Kim and colleagues. It presents a very interesting thickness-dependent Raman study of antiferromagnetic compound NiPS₃.

The main claim is that when reduced to a monolayer, this system provides a 2D realization of the XY model. This is of course an exciting statement, that should be strongly substantiated by the experimental data.

As pointed out by the authors, Raman scattering is one of the few available probes of magnetism in such reduced dimensions, and this is naturally the technique they used to support their claims.

Overall, I find the paper very interesting and believe that the experiments have been properly carried out. In fact, I would like to congratulate the authors for the quality of their experimental data, that have been obtained in a very systematic manner and provide a robust basis for discussion. The paper is well written, but I believe that some aspects of the presentation and discussion should be somewhat revised before considering this paper for publication in Nature Communications.

The disappearance of the long range magnetic order, which provides the main clue for the claim of the realization of XY physics, is convincingly supported by the disappearance of the two-magnon feature below 2UC thickness. However, quite clearly the structure of the 1UC is different from that of the thicker layers as evidenced by a strongly renormalized Raman spectra. This is not clearly shown nor discussed, but my feeling is that there are more features in the 1UC spectra than expected from the group theory analysis. Is the structure really the expected D3d? In fact, by looking at xy spectra shown in Fig. 3. the spectra look qualitatively different (e.g. new mode at 550cm⁻¹, structure around 200cm⁻¹) which is somewhat worrisome. Can the authors comment on this?

My main concern in the discussion concerns the quasi-elastic line, attributed without the slightest explanation to magnetic fluctuations. Should this be the case, why would those have a different symmetry than the 2 magnon peaks and show up in parallel polarization?

Shouldn't one rather naturally expect stronger elastic intensity in this case compared to the crossed polarization simply from the polarization dependence of the Rayleigh scattering term? Secondly, the discussion on the Bose correction that is provided at the end of the paper is highly unconvincing and the Fig. 6c resulting from it completely unreadable.

It would be instructive to show the Bose-corrected Raman spectra as function of temperature. Contrary to the claim of the authors, it rather seems to me like a trivial temperature dependence of this feature. It is also not clear what the origin of the peaks at low frequency and observed at low temperature in the bulk materials are.

Assuming this peak is indeed coming from the fluctuations, why shouldn't it get reinforced as the magnetic order presumably disappears in the single layer? So to me, this part of the discussion hardly makes sense and does not serve the case well. I would invite the authors to revise it considerably.

In conclusion I am left with a mixed feeling. The data are of great quality, but their presentation could be strongly improved. It is unclear to me that the single layer is indeed the same material as

the thicker ones, and I am therefore not completely convinced that the suppression of the two magnon peak bears the deep physics that the authors try to attribute to it. Finally, I find the entire discussion of the quasielastic peak simply misleading.

That being said, the set of data has a great potential, and with adequate revision (and maybe softer claims) the paper should be publishable in Nature Communications.

Reviewer #3 (Remarks to the Author):

This paper reports a systematic Raman scattering study of bulk crystals and exfoliated layers of the quasi-two-dimensional (quasi-2D) antiferromagnet NiPS₃. The authors observe a two-magnon feature that depends weakly on the layer thickness for samples with thicknesses down to two monolayers. For one-monolayer-thick samples, the peak becomes much weaker and can no longer be clearly identified. A related behavior is also observed for low-energy quasielastic fluctuations, and for a phonon anomaly indicating a structural transition coincident with the Neel transition. The authors interpret these observations in terms of the XXZ model, as a consequence of vortex-antivortex excitations that obliterate long-range magnetic order in the 2D limit.

As the two-magnon peak is primarily sensitive to nearest-neighbor spin correlations in the antiferromagnetic layers, the claim of an abrupt crossover in the intrinsic magnetism from two- to one-monolayer-thick samples is questionable a-priori. To substantiate their claim, the authors would have to conclusively rule out extrinsic factors such as differences in structural integrity or the presence of adsorbates between the two samples. This has not been accomplished. On the contrary, the phonon modes in the on-monolayer sample (Fig. 3) are much broader than those of the two-monolayer sample (Fig. S5), and extra modes appear to be present, indicating that such extrinsic factors may well be at the root of the crossover the authors have observed.

Even if the authors had conclusively demonstrated the intrinsic origin of the observed crossover, the interpretation in terms of the 2D XXZ model would be highly doubtful. The phonon anomalies in the bulk limit indicate strong magneto-structural coupling whose microscopic origin is far from obvious. Indeed, the spin structure displayed in Fig. 1 (with antiferromagnetic exchange coupling on some nearest-neighbor bonds, and ferromagnetic coupling on others) requires spin correlations that are not captured by such simple models. In such a situation, Ising-type spin-space anisotropies within the planes are generically expected, so the interpretation in terms of isotropic XY- or XXZ-type correlations is highly doubtful.

As the validity of the central claims in the manuscript is highly questionable, I recommend its rejection. The material at hand may well be interesting, but unraveling its microscopic magnetism will require more controlled experiments with more than a single experimental probe, and much more elaborate analysis and modelling than those that are applied in the current manuscript.

Authors' Response to the Reviewers' Comments:

We thank the reviewers for careful reading of our manuscript and valuable comments. Reviewer #1 commented that "*The paper is clearly written, and the scientific results taken at face value would be of sufficient interest to justify publication in Nature communication*" and asked us questions on "*potential artefacts or other possibilities that can affect these delicate measurements*" that we should address before she/he can recommend publication of their manuscript. Reviewer #2 commented that "*Overall, I find the paper very interesting and believe that the experiments have been properly carried out. In fact, I would like to congratulate the authors for the quality of their experimental data, that have been obtained in a very systematic manner and provide a robust basis for discussion. The paper is well written.*" Reviewer #2 also believes that "*some aspects of the presentation and discussion should be somewhat revised before considering this paper for publication in Nature Communications*" and concludes that "*the set of data has a great potential, and with adequate revision (and maybe softer claims) the paper should be publishable in Nature Communications.*" Finally, Reviewer #3 commented that "*The material at hand may well be interesting, but unraveling its microscopic magnetism will require more controlled experiments with more than a single experimental probe, and much more elaborate analysis and modelling than those that are applied in the current manuscript*" and provided very important and useful suggestions.

In order to address their concerns, we have performed a significant amount of new experiments and theoretical analyses and also have repeated some measurements with much longer integration times to obtain results with higher S/N ratios. Thanks to

our additional efforts in responding to the requests from the reviewers, our conclusion is now based on a much better foundation.

There are many issues on our manuscript that have been pointed out by all three reviewers. Considering that all three reviewers are the best experts in the field, we believe that those commonly raised issues are of utmost importance. Therefore, in the following, we will first discuss the results of our new experiments and theoretical analyses to address the issues raised by all three reviewers, and then list our point-by-point response to each of the reviewers' comments.

Response to Common Comments by All Reviewers

1. Improved estimation of the Néel temperature

First, we repeated the temperature dependence of the Raman spectra with new samples on different substrates (SiO_2 and hBN) and using an improved filter for a better S/N ratio. Moreover, using a much longer integration time also helped to improve the S/N ratio, especially for thinner samples (1L or 2L). The Néel temperature was then determined more systematically by analyzing the splitting of the phonon P_2 with the use of a spin-induced phonon frequency shift model. The new data are presented in the revised Fig. 5 and the data analysis procedures are now explained in Supplementary Note 3 and Supplementary Fig. 12. These new analyses revealed a previously unresolved small decrease of the Néel temperature for 2L and 3L and confirmed our previous main conclusion that the magnetic phase transition is significantly suppressed for 1L.

Relevant Revisions

- **Revised Fig. 5 and Supplementary Fig. 11**
- **Added Supplementary Note 3 and Supplementary Fig. 12**
- **Changes in manuscript (Page 13, line 3)**

Now, we would like to inspect this splitting as a function of temperature for few-layer NiPS₃ (see Supplementary Fig. S7). Figure 5 summarizes the results for several thicknesses including monolayer. As seen in the figure, the splitting shows a clear onset for 2-layer or thicker samples near T_N of the bulk. On the contrary, the splitting for monolayer NiPS₃, if any, is virtually temperature independent. These results confirm our conclusion from the analysis of the two-magnon signals: (1) the antiferromagnetic transition temperature (T_N) is almost independent of thickness for 2-layer or thicker NiPS₃ and (2) antiferromagnetic ordering even at very low temperature (around 10 K) is suppressed in monolayer NiPS₃

→ Now, we would like to inspect this splitting as a function of temperature for few-layer NiPS₃ (see Supplementary Fig. 11). Figure 5a shows temperature dependence of P₂ splitting for various thicknesses including monolayer. The antiferromagnetic transition temperatures are extracted by using the spin-induced phonon frequency shift model^{34,35} (see Supplementary Note 3 and Supplementary Fig. 12 for details) and summarized in Fig. 5b. As seen in Fig. 5a, the splitting shows a clear onset for 2-layer or thicker samples and the transition temperature gets slightly lower as the thickness decreases. On the contrary, the splitting for monolayer NiPS₃, if any, is virtually temperature independent. These results confirm our conclusion from the analysis of the two-magnon signals: (1) the antiferromagnetic transition temperature (T_N) depends only slightly on the thickness for 2-layer or thicker NiPS₃, and (2) antiferromagnetic ordering is significantly suppressed in monolayer NiPS₃.

2. Improved analysis of the temperature dependence of the quasi-elastic scattering (QES) signal

We found that the analysis of the temperature dependence of the quasi-elastic scattering (QES) signal contained some errors in the previous version. By studying previous publications in the literature, we have derived expressions for the relevant spectral weight of χ''/ω , which is related to the magnetic specific heat, in Eqs. (3) and (4) and analyzed the data accordingly. The improved data for 1L, 2L, 3L, and 9L are now summarized in revised Fig. 6 and Supplementary Fig. 15. The comparison clearly demonstrates how 1L behaves differently from other thicknesses and supports our conclusion that the antiferromagnetic ordering is suppressed for 1L.

Relevant Revisions

- Revised Fig. 6 and Supplementary Fig. 15
- Changes in manuscript (Page 14, line 12)

We further analyze the low-frequency QES signals due to the magnetic fluctuations^{21,26}. Since such QES signals are suppressed in the presence of the spin ordering in ferromagnetic or antiferromagnetic materials, its suppression can also be used as another good indicator for magnetic ordering. Figures 6a and 6b compare the temperature dependence of the low-frequency region of the Raman spectra of bulk and monolayer NiPS₃. Whereas the QES signals from bulk NiPS₃ are strongly suppressed below T_N , a considerable amount of spectral weight persists down to the lowest temperature in the case of monolayer NiPS₃. On the other hand, samples with intermediate thicknesses show the trend similar to that of bulk NiPS₃ (see Supplementary Figs S5). For a further quantitative analysis, the temperature dependence of the phonon population should be considered. The Raman response for

Stokes scattering excluding the spurious effect of the phonon occupation factor is given by

$$\chi''(\omega) = S(\omega) / [1 + n(\omega)], \quad (3)$$

where $S(\omega)$ is the measured intensity and $n(\omega) = [\exp(\hbar\omega/k_B T) - 1]^{-1}$ is the Bose-Einstein factor. Likewise, the renormalized responses for anti-Stokes scattering is given by

$$\chi''(\omega) = S(\omega) / n(\omega). \quad (4)$$

We obtained the renormalized spectral weight of QES by integrating the corrected Raman response in the range 10–50 cm^{-1} . Figure 6c shows the temperature dependence of QES Raman intensities observed in both parallel- and cross-polarization configurations for several thicknesses. In order to compare the temperature dependences accurately, we normalized the data for each thickness by the maximum value of $\chi''(\omega)$ for the chosen thickness (see Supplementary Fig. S8). According to this analysis, for all thicknesses except monolayer, the renormalized spectral weight of QES is largest near bulk T_N and becomes smaller as the temperature decreases therefrom due to magnetic ordering. On the other hand, in the case of monolayer NiPS_3 , the renormalized spectral weight of QES almost monotonically increases as the temperature is lowered. These results again support our interpretation that antiferromagnetic ordering is drastically suppressed in monolayer NiPS_3 . The increase in the spectral weight of QES at low temperatures indicates that spin fluctuations in monolayer NiPS_3 ,

→ We further analyze the low-frequency QES signals due to the magnetic fluctuations^{21,24}. Since such QES signals are suppressed in the presence of the spin ordering in ferromagnetic or antiferromagnetic materials, its suppression can also be

used as another good indicator for magnetic ordering. Figures 6a and 6b compare the temperature dependence of the low-frequency region of the polarized Raman spectra of 9L and monolayer NiPS₃. In parallel polarization configuration, the QES signals from 9L NiPS₃ are strongly suppressed below T_N , but a considerable amount of spectral weight persists down to the lowest temperature in the case of monolayer NiPS₃. The difference is even more striking for the case of cross polarization configuration. Here, the QES signals from 9L NiPS₃ are somewhat enhanced near T_N , presumably because of strong spin fluctuations near the phase transition²⁵. On the other hand, the QES signals from monolayer NiPS₃ monotonically increase as the temperature is lowered as if it is approaching the phase transition. The temperature dependences of QES signals for intermediate thicknesses, such as 2L and 3L NiPS₃, are qualitatively similar to that of 9L NiPS₃ (see Supplementary Fig. 15). For a more quantitative analysis, the temperature dependence of the phonon population should be considered. The measured intensity of Stokes-shifted QES is simply expressed by^{35,40-42}

$$I(\omega) \propto \frac{\omega}{1 - e^{-\hbar\omega/k_B T}} \frac{C_m T D k^2}{\omega^2 + (Dk^2)^2}, \quad (3)$$

where C_m is the magnetic specific heat and D is the thermal diffusion constant $D = K / C_m$ with the magnetic contribution to the thermal conductivity K . Since the Raman response $\chi''(\omega)$ for Stokes scattering is given by $\chi''(\omega) = I(\omega) / [n(\omega) + 1]$, where $n(\omega) = [\exp(\hbar\omega / k_B T) - 1]^{-1}$ is the Bose-Einstein factor, Eq. (3) is expressed in terms of the Raman response $\chi''(\omega)$ ^{35,42},

$$\frac{\chi''(\omega)}{\omega} \propto C_m T \frac{Dk^2}{\omega^2 + (Dk^2)^2}. \quad (4)$$

We obtained the spectral weight of QES by integrating the normalized Raman response χ''/ω in the range of 11–40 cm^{-1} . Figure 6c shows the temperature dependence of the spectral weight of QES in both parallel- and cross-polarization configurations for several thicknesses. For both polarization configurations, the spectral weight of QES shows a peak near the magnetic transition temperature for all thicknesses except monolayer. In the case of monolayer NiPS₃, the spectral weight of QES almost monotonically increases as the temperature is lowered in both polarization configurations as if it is approaching the phase transition from above. These results are consistent with our prior observation from the raw Raman data in Figs. 6a, b and support our interpretation that antiferromagnetic ordering is drastically suppressed in monolayer NiPS₃. The increase in the spectral weight of QES at low temperatures indicates that spin fluctuations in monolayer NiPS₃,

3. Explanation for the origin of new peaks appearing in thin layers

All three reviewers expressed concerns about a broad and strong peak (P_3) near 210 cm^{-1} , which is absent in bulk but appears in the spectrum of monolayer NiPS₃. Because of the presence of this peak, they wondered whether the monolayer sample was damaged by degradation or the observed suppression of the magnetic ordering in monolayer is due to some extrinsic effects. We suspected that this peak is due to resonance-enhanced multiphonon scattering that is frequently observed in many 2-dimensional materials. For example, in MoS₂, the signal from 2-phonon scattering of zone-boundary longitudinal acoustic phonons (2LA) is strongly enhanced for resonant excitation of 1.96 eV and dominates the spectrum, with an intensity much stronger than the main Raman-active zone-center optical phonon modes. This phenomenon has been explained in terms of the interplay between the large

densities of states of the zone-boundary phonons and the electronic bands that are resonant with the excitation laser [35]. In order to verify this interpretation, we carried out additional Raman measurements on 1-3L and bulk NiPS₃ samples using several lasers. As explained in Supplementary Note 2 and Supplementary Fig. 9, P₃ is present in 1-3L but absent in bulk NiPS₃ when the 2.41-eV excitation is used. When the excitation energy is slightly increased to 2.54 eV, P₃ disappears for 2L and 3L and is significantly decreased for 1L. The intensities of the other peaks also decrease, indicating that we are moving away from the resonance, but P₃ is preferentially suppressed, supporting our hypothesis that this peak is preferentially enhanced due to a special resonance condition. For the 2.81-eV excitation, P₃ is completely suppressed for all samples, but other peaks are also greatly diminished. In our original work, we chose to use the 2.41-eV excitation because it gave the largest signal for the main features in the Raman spectrum. At the time, we did not pay much attention to P₃ because similar features had been frequently observed in our work on other 2-dimensional materials. We further checked the temperature dependence of P₃ for 1-4L NiPS₃ (see Supplementary Fig. 10). Since the frequency and the intensity of P₃ do not exhibit any significant change near the Néel temperature, we can safely conclude that P₃ is not related to the magnetic ordering. Thanks to the comments by the reviewers, we have been able to positively identify the origin of this peak and safely confirm that it does not affect our conclusion.

Relevant Revisions

- Added Supplementary Note 2 and Supplementary Figs. 8-10
- Added in manuscript (Page 12, Line 18)

→ We note that there are several peaks that originate from multi-phonon scattering at ~210, ~590, and ~800 cm⁻¹ (see Supplementary Fig. 8). Some of these peaks are

strongly enhanced due to the resonance effect^{34,35}. For example, the relative intensities or the line shapes of P_3 at $\sim 210\text{ cm}^{-1}$ or the peak at $\sim 590\text{ cm}^{-1}$ vary with the excitation laser energy, which is a clear indication of resonant processes (see Supplementary Note 2 and Supplementary Fig. 9). We further checked the temperature dependence of P_3 for 1-4L NiPS₃ (see Supplementary Fig. 10). Since the frequency and the intensity of P_3 does not exhibit any significant change near the Néel temperature, we can safely conclude that P_3 is not related to the magnetic ordering.

4. Measurement on NiPS₃ on hBN (examination of extrinsic effects)

In order to examine the possibility of the SiO₂/Si substrate influencing the magnetic ordering in thin NiPS₃, we fabricated new NiPS₃ samples on hexagonal boron nitrides (hBN). Since hBN is known to eliminate many extrinsic effects caused by the SiO₂/Si substrate such as doping or charged impurity scattering, we believe this is the best test except for possibly measuring a suspended sample which is almost impractical. Remarkably, as shown in Supplementary Figs. 13 and 14, none of the major features such as the P_2 splitting, the 2-magnon scattering signal, the Fano resonance of P_9 , and the QES signals show any discernible difference between the results for samples on hBN and those for samples on SiO₂/Si. Therefore, we can safely conclude that none of the observed magnetic ordering or lack thereof in few-layer NiPS₃ was influenced by the extrinsic substrate effects in any significant manner, and our observation of the suppression of the antiferromagnetic ordering in 1L is an intrinsic property of the material.

Relevant Revisions

Added in manuscript (Page 14, line 7)

→ In order to rule out the possible extrinsic effect due to the substrate, we fabricated additional samples on hexagonal boron nitride (hBN) layers on SiO₂/Si substrates. We found that there is virtually no difference between the results for samples exfoliated directly on SiO₂/Si and those for samples on hBN/SiO₂/Si (see Supplementary Figs. 13 and 14), which strongly indicate that the extrinsic effect due to the substrate is not important.

Now we present our point-by-point response to each of the reviewers' comments

Reviewer #1

Comments to the authors

The paper reports the polarized Raman study of the thickness dependent magnetic order in a van der Waals material, NiPS₃, which is believed to be an XXZ-type antiferromagnet. They found that the Raman spectra changed dramatically when NiPS₃ is thinned down to monolayer. Together with the theoretical calculation, the paper claims that the antimagnetic order of NiPS₃ becomes unstable in the monolayer limit. The paper is clearly written, and the scientific results taken at face value would be of sufficient interest to justify publication in Nature communication.

Authors' Response

We thank the reviewer for careful reading of our manuscript and encouraging comments.

Comments to the authors 1

How did the authors confirm the quality of the bulk crystal? How is the $T_n=155$ K extracted?

Authors' Response

We checked the orientation and the quality of individual bulk crystal using a single crystal X-ray diffractometer (XtaLAB P200, Rigaku). Our data showed clear Bragg peaks for all directions, confirming the high quality of our NiPS₃ samples (See

Supplementary Fig. 17). In addition, our $d\chi/dT$ curve obtained from a commercial magnetometer (MPMS) clearly showed the sharp peaks for both directions (See Supplementary Fig. 2), where T_N corresponds to the peak of $d\chi/dT$ (~ 155 K). The T_N of few-layer samples were determined by analyzing the energy splitting of P_2 by using the spin-induced phonon frequency shift model as we explained before (1. Improved estimation of the Néel temperature).

Relevant Revisions

- **Revised Supplementary Fig. 2**
- **Added Supplementary Fig. 17**

Comments to the authors 2

The Raman spectrum is almost the same as bulk down to 2L and suddenly changed at 1L, instead of a gradual evolve, which is somewhat unexpected since 2L is already quite 2D. Do the authors have any explanation about this? Could the dramatic change of Raman spectrum of 1L NiPS₃ be due to degradation of the flake? The authors should do additional experiments to confirm the stability of 1L NiPS₃ and exclude this possibility.

Authors' Response

All our samples were kept in vacuum immediately after exfoliation and throughout measurements. In order to check the possibility of degradation, we compared the atomic force microscopy images of the samples before and after measurements in various conditions. It turned out that significant photo-degradation was observed when the measurements were carried out in air, but no such damage was seen when the measurements were done in vacuum using a laser power of 100 μ W as in our work. At higher power, slight damage was observed even when the

measurements were done in vacuum. This information is added in Supplementary Note 5 and Supplementary Fig. 18. Therefore, we can conclude that there is no significant degradation in our samples that will influence the major scientific conclusion of the work.

In order to answer the reviewer's question regarding the difference between 1L and 2L cases, we would like to point out that although 2L is already quite 2D-like, the significant difference is the existence of interlayer interaction in 2L. Although a rigorous explanation would require a highly sophisticated theory, our Monte Carlo simulation qualitatively explains our observation. Our new, expanded Monte Carlo simulation results are presented in Supplementary Note 4 and Supplementary Fig. 16. According to our simulations, the Néel temperature slowly decreases as the thickness is reduced down to 2L and then decreases dramatically for 1L. This result is consistent with our experimental observation.

Relevant Revisions

- **Added Supplementary Note 5 and Supplementary Fig. 18**
- **Revised Supplementary Note 4 and Supplementary Fig. 16**
- **Changes in manuscript (page 16, line 3)**

In order to shed further light on the experimental observations, we have also carried out Monte Carlo simulations as a function of the number of layers (see Supplementary Figs. S9), and the results support our interpretation of the experimental data: there are qualitative differences in our Monte Carlo simulation results between monolayer and thicker systems.

➔ In order to shed further light on the experimental observations, we have carried out Monte Carlo simulations as a function of the number of layers (see Supplementary Note 4 and Supplementary Fig. 16). The simulation results qualitatively support our

interpretation of the experimental data. The simulated Néel temperature slowly decreases as the thickness is reduced, down to 2L case, but the decrease is more dramatic for monolayer. For a better quantitative agreement, more sophisticated calculations will be necessary, which is beyond the scope of this work.

Comments to the authors 3

For the Raman spectra of 1L NiPS₃ (Fig. 3b and Fig. S7f), some Raman peaks (~200, ~600, ~800 cm⁻¹ in cross-polarization configuration and ~210, ~250 cm⁻¹ in the parallel- polarization configuration) still show quite strong temperature dependence. Do the authors know the origin of this Raman modes and their temperature dependences? This seems inconsistent with the conclusion of the suppressed magnetic order at 1L limit.

Authors' Response

As for the broad peak P₃ near 210 cm⁻¹, please refer to our previous explanation (3. Explanation for the origin of new peaks appearing in thin layers).

The peak(s) near 250 cm⁻¹ that appears in parallel polarization configuration and only at low temperatures is present in samples with any thickness and slowly redshifts as the thickness is reduced. Although it is somewhat broader in 1L case, the temperature dependence is similar in cases. Since there are several Raman modes in the vicinity of this peak, some modes might meet a special resonance condition to cause the broadening in 1L as in the case of P₃. In any event, this peak does not show much correlation with the Néel temperature and so should not affect our main conclusion.

The signal near 600 cm⁻¹ is a combination of a sharp A_g mode (P₁₀) and multi-phonon scattering signals as clearly seen in Supplementary Fig. 8. Since the A_g

mode should be suppressed in cross polarization configuration, the observed signal must be mostly from multi-phonon scattering. In our DFT calculation, there is no phonon mode with a frequency higher than the A_g mode (559.25 cm^{-1}) which corresponds to P_{10} with the experimental frequency of 595 cm^{-1} . We believe that the multi-phonon scattering is resonantly enhanced as in the case of P_3 because its relative intensity and the line shape vary with the energy of the excitation laser as shown in Supplementary Fig. 9.

Similarly, the signal near 800 cm^{-1} present in all samples with different thickness is also a multi-phonon scattering feature.

Relevant Revisions

- Added Supplementary Figs. 8 and 9
- Added Supplementary Note 2
- Added text description (page 12, line 18)

→ We note that there are several peaks that originate from multi-phonon scattering at ~ 210 , ~ 590 , and $\sim 800 \text{ cm}^{-1}$ (see Supplementary Fig. 8). Some of these peaks are strongly enhanced due to the resonance effect. For example, the relative intensities or the line shapes of P_3 at $\sim 210 \text{ cm}^{-1}$ or the peak at $\sim 590 \text{ cm}^{-1}$ vary with the excitation laser energy, which is a clear indication of resonant processes (see Supplementary Note 2 and Supplementary Fig. 9).

Comments to the authors 4

Data quality of 1L NiPS3 is not that satisfactory, especially for these delicate measurements. Could the authors try to improve signal-to-noise ratio by, for example, increase integration time?

Authors' Response

We thank Reviewer #1 for this important suggestion. We carried out new temperature-dependent polarized Raman measurements using longer integration times and with a new improved filter. Consequently, the S/N ratio of the data has improved greatly as shown in Figs. 3, 6, and Supplementary Fig. 11. This has made it possible for us to resolve a small decrease in T_N of 2L and 3L NiPS₃ as we explained before (1. Improved estimation of the Néel temperature).

Relevant Revisions

- **Revised Figs. 3 and 6 and Supplementary Fig. 11**
- **Revised analyses based on these data**

Reviewer #2

Comments to the authors

I have read with great attention the paper by Kangwon Kim and colleagues. It presents a very interesting thickness-dependent Raman study of antiferromagnetic compound NiPS₃.

The main claim is that when reduced to a monolayer, this system provides a 2D realization of the XY model. This is of course an exciting statement, that should be strongly substantiated by the experimental data.

As pointed out by the authors, Raman scattering is one of the few available probes of magnetism in such reduced dimensions, and this is naturally the technique they used to support their claims.

Overall, I find the paper very interesting and believe that the experiments have been properly carried out. In fact, I would like to congratulate the authors for the quality of their experimental data that have been obtained in a very systematic manner and

provide a robust basis for discussion. The paper is well written, but I believe that some aspects of the presentation and discussion should be somewhat revised before considering this paper for publication in Nature Communications.

Authors' Response

We thank the reviewer for the positive evaluation of our work.

Comments to the authors

The disappearance of the long range magnetic order, which provides the main clue for the claim of the realization of XY physics, is convincingly supported by the disappearance of the two-magnon feature below 2UC thickness. However, quite clearly the structure of the 1UC is different from that of the thicker layers as evidenced by a strongly renormalized Raman spectra. This is not clearly shown nor discussed, but my feeling is that there are more features in the 1UC spectra than expected from the group theory analysis. Is the structure really the expected D3d? In fact, by looking at xy spectra shown in Fig. 3. the spectra look qualitatively different (e.g. new mode at 550cm⁻¹, structure around 200cm⁻¹) which is somewhat worrisome. Can the authors comment on this?

Authors' Response

As we explained in our response to Comment 3 by Reviewer #1 and in our response to All Reviewers '3. Explanation for the origin of new peaks appearing in thin layers', most of the 'new' peaks are due to resonantly enhanced multi-phonon scattering in few layer NiPS₃. Furthermore, we believe that symmetry group change in 1L is not responsible for the differences in the Raman spectrum because the changes in the Raman spectrum occurs gradually as the thickness is reduced. These are clearly seen in the improved Raman spectra presented in Supplementary Figs. 8 and 9.

Relevant Revisions

- Added Supplementary Figs. 8 and 9
- Added Supplementary Note 2

Comments to the authors

My main concern in the discussion concerns the quasi-elastic line, attributed without the slightest explanation to magnetic fluctuations. Should this be the case, why would those have a different symmetry than the 2 magnon peaks and show up in parallel polarization? Shouldn't one rather naturally expect stronger elastic intensity in this case compared to the crossed polarization simply from the polarization dependence of the Rayleigh scattering term?

Authors' Response

The quasi-elastic scattering (QES) signal is observed occasionally in Raman scattering of magnetic materials, and to the best of our knowledge, its exact mechanism is not fully understood yet. One can qualitatively explain that it is due to low-energy magnetic fluctuations in the system. Since it involves only low energy spin fluctuations, it is reasonable to assume that the polarization of the scattered light would not be much different from that of the incident light, similar to the case of Rayleigh scattering. This is what is observed. However, in the lack of a credible theory, we are reluctant to comment on the scattering mechanism. In the revised version, we added a description on the polarization behavior of the QES signal.

Relevant Revisions

Changes in manuscript (page 7, line 10)

Also, the clear signals centered at 0 cm^{-1} in the spectrum obtained in the parallel-polarization configuration at $T=295 \text{ K}$ are due to quasi-elastic scattering (QES) from

magnetic fluctuations. These QES signals become considerably weakened at lower temperatures since the spin fluctuations are suppressed as the antiferromagnetic ordering gets developed. Such QES signals are often observed in low-dimensional spin systems^{21,27,28}.

→ Also, the clear signals centered at 0 cm^{-1} in the spectrum obtained in the parallel-polarization configuration at $T=295 \text{ K}$ are ascribed to quasi-elastic scattering (QES) from magnetic fluctuations, which is often observed in low-dimensional spin systems^{21,25,26}. These QES signals become considerably weakened at lower temperatures since the spin fluctuations are suppressed in the magnetically ordered phase. Unlike the two-magnon signal which does not depend on the polarization of the scattered light, the QES signal is much stronger in parallel polarization configuration, presumably because low-energy spin fluctuations do not change the polarization of the scattered photon very much.

Comments to the authors

Secondly, the discussion on the Bose correction that is provided at the end of the paper is highly unconvincing and the Fig. 6c resulting from it completely unreadable.

It would be instructive to show the Bose-corrected Raman spectra as function of temperature. Contrary to the claim of the authors, it rather seems to me like a trivial temperature dependence of this feature. It is also not clear what the origin of the peaks at low frequency and observed at low temperature in the bulk materials are.

Assuming this peak is indeed coming from the fluctuations, why shouldn't it get reinforced as the magnetic order presumably disappears in the single layer? So to me, this part of the discussion hardly makes sense and does not serve the case well.

I would invite the authors to revise it considerably.

Authors' Response

We thank the reviewer for pointing out the discrepancy in our analysis. We reviewed several papers on similar subjects and found that indeed our original analysis was not quite correct and that in the original manuscript we did not include a thorough description, with proper reference, of the analysis procedure. In the revised version, we therefore present both the original (unprocessed) Raman spectra and the renormalized Raman response χ''/ω in Fig. 6 and Supplementary Fig. 15. For more details, please refer to our response to all Reviewers above [2. Improved analysis of the temperature dependence of the quasi-elastic scattering (QES) signal].

Relevant Revisions

- **Revised Fig. 6 and Supplementary Fig. 15**

Comments to the authors

In conclusion I am left with a mixed feeling. The data are of great quality, but their presentation could be strongly improved. It is unclear to me that the single layer is indeed the same material as the thicker ones, and I am therefore not completely convinced that the suppression of the two magnon peak bears the deep physics that the authors try to attribute to it. Finally, I find the entire discussion of the quasielastic peak simply misleading.

That being said, the set of data has a great potential, and with adequate revision (and maybe softer claims) the paper should be publishable in Nature Communications.

Authors' Response

Again, we thank the reviewer for the positive encouragement. With the revisions, we believe our conclusions are on more solid ground.

Reviewer #3

Comments to the authors

This paper reports a systematic Raman scattering study of bulk crystals and exfoliated layers of the quasi-two-dimensional (quasi-2D) antiferromagnet NiPS₃. The authors observe a two-magnon feature that depends weakly on the layer thickness for samples with thicknesses down to two monolayers. For one-monolayer-thick samples, the peak becomes much weaker and can no longer be clearly identified. A related behavior is also observed for low-energy quasielastic fluctuations, and for a phonon anomaly indicating a structural transition coincident with the Neel transition. The authors interpret these observations in terms of the XXZ model, as a consequence of vortex-antivortex excitations that obliterate long-range magnetic order in the 2D limit.

Authors' Response

We thank the reviewer for careful reading of our manuscript and for important criticisms and useful suggestions.

Comments to the authors

As the two-magnon peak is primarily sensitive to nearest-neighbor spin correlations in the antiferromagnetic layers, the claim of an abrupt crossover in the intrinsic magnetism from two- to one-monolayer-thick samples is questionable a-priori. To substantiate their claim, the authors would have to conclusively rule out extrinsic factors such as differences in structural integrity or the presence of adsorbates between the two samples. This has not been accomplished. On the contrary, the phonon modes in the one-monolayer sample (Fig. 3) are much broader than those of

the two-monolayer sample (Fig. S5), and extra modes appear to be present, indicating that such extrinsic factors may well be at the root of the crossover the authors have observed.

Authors' Response

It is reasonable to suspect extrinsic effect when some unusual effects are observed experimentally. In response to the reviewers' comments, we have carried out several new, control measurements to rule out the possibility of extrinsic effects. As we explained in our response to all Reviewers [4. Measurement on NiPS₃ on hBN (examination of extrinsic effects)] and in our response to Comment 2 of Reviewer #1, our control measurements indicate that extrinsic effects, if any, are not strong enough to affect the major findings of our work. Furthermore, the apparent extra modes in 1L samples are explained in our response to all Reviewers (3. Explanation for the origin of new peaks appearing in thin layers) and in our response to Comment 3 of Reviewer #1.

Comments to the authors

Even if the authors had conclusively demonstrated the intrinsic origin of the observed crossover, the interpretation in terms of the 2D XXZ model would be highly doubtful. The phonon anomalies in the bulk limit indicate strong magneto-structural coupling whose microscopic origin is far from obvious. Indeed, the spin structure displayed in Fig. 1 (with antiferromagnetic exchange coupling on some nearest-neighbor bonds, and ferromagnetic coupling on others) requires spin correlations that are not captured by such simple models. In such a situation, Ising-type spin-space anisotropies within the planes are generically expected, so the interpretation in terms of isotropic XY- or XXZ-type correlations is highly doubtful.

Authors' Response

We thank the reviewer for the thoughtful comments. The reviewer has a point in that NiPS₃ is not an ideal XY or XXZ system. Some degree of in-plane anisotropy indeed exists. However, the important difference from the Ising system is the continuous in-plane degree of freedom in our case. This is a clear difference from the Ising systems such as FePS₃ [8]. The experimental observations also support this view because the antiferromagnetic ordering survived down to the 1L limit in the Ising system of FePS₃, whereas it is suppressed in the current work.

As the reviewer points out, the microscopic origin of the strong magneto-structural coupling is not easily understood. But we would like to note that magneto-elastic coupling in a similar van der Waals material Cr₂Ge₂Te₆ was reported by Y. Tian and K. Burch *et al.* [33]. They found several results from the strong exchange-striction type spin-phonon coupling present in this material: low energy mode splitting, magnetic quasi-elastic scattering in the paramagnetic phase, and a clear upturn and line broadening of phonon modes below T_c. We have added this work as a reference in the revised version although there may be some differences in the exact mechanism between these materials.

As for the spin structure in Fig. 1, we would like to mention that there are numerous theoretical studies on the phase diagram of the J₁-J₂-J₃ XXZ model on the honeycomb lattice, for both classical and quantum spin systems [Supplementary Refs. 5-8]. The essence of these studies is that the zig-zag magnetic order can be stabilized by the sufficiently large J₂ and J₃, which was confirmed in the case of Na₂IrO₃ from the spin wave dispersion obtained by inelastic neutron scattering measurements [PRL 108, 127204 (2013)]. According to the recent powder inelastic neutron scattering results (arXiv:1808.01480), the non-zero gap of the spin wave in

NiPS₃ is small (~7 meV), which means that the single-ion anisotropy is very small. This is a clear contrast to the case of Ising-type FePS₃ in which the easy-axis anisotropy is much larger than J_1 [Supplementary Ref. 10]. The dimensionality of the system can also be estimated from the critical behaviors. According to Ref. [16], the critical exponent $\beta \sim 0.3$ near T_N , showing a 3 dimensional correlation, but $\beta = 0.15$ below approximately $0.9T_N$, exhibiting a crossover to a 2-dimensional correlation ($0.1 \leq \beta \leq 0.25$ corresponds to a 2-dimensional system). All these arguments can be summarized that it is reasonable to describe NiPS₃ at low temperatures as an XY-like or XXZ-like system.

Nevertheless, we agree that the readers might get confused, and therefore we revised the manuscript to remove ambiguities.

Relevant Revisions

- Added text (page 5, line 19)

→ We note that due to the ab anisotropy, this is not an exact XXZ system, but an approximate one.

- Added text and ref. (page 12, line 16)

→ We note that a similar phonon splitting below critical temperature³³ has been reported for Cr₂Ge₂Te₆.

Comments to the authors

As the validity of the central claims in the manuscript is highly questionable, I recommend its rejection. The material at hand may well be interesting, but unraveling its microscopic magnetism will require more controlled experiments with more than a single experimental probe, and much more elaborate analysis and modelling than those that are applied in the current manuscript.

Authors' Response

Taking seriously the concerns and criticisms of Reviewer 3 into account, we have performed a significant amount of new experiments and analyses as we explained at the beginning of this response. We are very thankful to the reviewer for helping us to remarkably improve our paper. One comment however that we were not capable of following was to do more controlled experiments using other experimental methods than Raman spectroscopy. Although usual bulk materials can be studied by other experimental probes, it is not the case for the atomically thin 2-dimensional materials. As Reviewer #2 commented, "*Raman scattering is one of the few available probes of magnetism in such reduced dimensions.*" Especially for antiferromagnetic materials, the magneto-optical Kerr effect (MOKE) that is useful for ferromagnetic 2-dimensional materials is not applicable. Until a more powerful probe is developed for the study of two-dimensional antiferromagnetic ~~van der Waals~~ materials, what we have done with Raman scattering will remain the best and the state-of-the-art experiments one can do with this highly interesting material.

Reviewers' comments:

Reviewer #1 (Remarks to the Author):

I appreciate the authors' efforts for performing additional experiments in response to my comments. Characterization of the bulk crystals shows reasonable quality. The S/N ratio of the data has been improved. The AFM measurements suggest no significant degradation in 1L samples, although I am not fully convinced by the multi-phonon scattering origin of the new peaks appearing in 1L NiPS₃. The interpretation of the quasi-elastic scattering signal is also not that satisfactory. Nevertheless, in my opinion, it does not undermine the major findings of the manuscript.

I am satisfied with the changes made to the manuscript and would like to recommend it to be published on Nature Communications.

Reviewer #2 (Remarks to the Author):

I have read the revised version of the paper by Cheong et al.

I am overall happy with the proposed revisions. I find the new data and the presentation strongly improved and find the present version of the manuscript very convincing.

I do recommend its publication in Nature Communications.

The only minor comment I would have left concerns the terminology 'P2' and 'splitting' for this mode. I find this a bit misleading.

There are in fact two modes, one with A_g and one with B_g symmetry that happen to have the same energy in the non-magnetic state, and a different one in the magnetic state.

I find the discussion very convincing, but the wording 'splitting' is more reminiscent of the degeneracy lifting of a mode, and this is not the case here.

So I'd rather refer to a 'magnetic-order-induced energy difference' or to some extra hardening or so than to a 'splitting'.

Reviewer #3 (Remarks to the Author):

The authors have made a valiant effort to clarify the issues that have been raised in the review. The paper now provides a fairly clear view of the capabilities and limitations of Raman scattering for the investigation of antiferromagnetic 2D materials. Nonetheless, publication in Nature Communications is not justified in my opinion, for the following reasons.

First, the Raman spectrum of the one-monolayer sample looks very different from those of thicker samples. The primary phonons are significantly broader, and there are several extra modes. The authors assert that some of the extra modes may be multiphonon excitations, and some indications of extra phonon modes are already visible in two- and three-monolayer samples. On balance, however, the evidence points to a substantial modification of the lattice structure and/or structural domain distribution in the thinnest samples. Especially in view of the strong magneto-elastic coupling, it is implausible to assume that this structural modification and the apparent modification of the magnetic properties are unrelated, and to attribute the degradation of the Neel temperature entirely to spin fluctuations. The central message of this paper is therefore dubious.

Second, the current study does not open up significant new perspectives for the investigation of antiferromagnetic 2D materials. Two-magnon Raman scattering probes high-energy magnetic fluctuations that are only indirectly related to the antiferromagnetic order parameter. Direct probes of antiferromagnetic order are not in sight, as the authors themselves have pointed out. It is not apparent how studies of this kind can uncover new information on XY or XXZ systems that is not already known from previous work on this subject. The authors' justification for publication in Nature Communications in the abstract ("This work provides the first experimental examination of the XY magnetism of two-dimensional systems.") is inappropriate as there has been extensive prior work on a variety of platforms ranging from ultrathin magnetic films to Josephson junction arrays. For a summary, see for instance Taroni et al., *J. Phys.: Condens. Matter* **20**, 275233 (2008). Currently there is no discussion at all of prior research on XY systems. A discussion of the context should be added, wherever the paper may eventually be published.

Authors' Response to the Reviewers' Comments:

Reviewer #1

Comments to the authors

I appreciate the authors' efforts for performing additional experiments in response to my comments. Characterization of the bulk crystals shows reasonable quality. The S/N ratio of the data has been improved. The AFM measurements suggest no significant degradation in 1L samples, although I am not fully convinced by the multi-phonon scattering origin of the new peaks appearing in 1L NiPS₃. The interpretation of the quasi-elastic scattering signal is also not that satisfactory. Nevertheless, in my opinion, it does not undermine the major findings of the manuscript.

I am satisfied with the changes made to the manuscript and would like to recommend it to be published on Nature Communications.

Authors' Response

We thank the reviewer for the careful reading of our manuscript and the encouraging comments and recommendation for publication.

Although the reviewer is now satisfied with our manuscript, we would like to elaborate more on the origin of the new peaks in 1L samples. In previous resonance Raman studies on 2-dimensional transition metal dichalcogenide materials such as MoS₂ and WS₂, several new peaks with large intensities were observed when the excitation energy was close to the exciton energies. For example, in MoS₂, the 2LA (two-longitudinal-acoustic-phonon scattering) signal is very weak if the excitation energy is off-resonance but becomes much stronger than the main one-phonon

scattering peaks if the resonant excitation energy of 1.96 eV is used (see Figure 1 below). The calculated phonon dispersion confirms that the frequency of this peak corresponds to twice the frequency of the zone-boundary phonon connected to the LA branch. As explained in Ref. [38], the large phonon density of states at the energy of the van Hove singularity due to these zone boundary phonons and the strong resonance effect combine to give strong Raman signal for this otherwise weak scattering. In WS₂, the 2LA signal is very close to the zone-center one-phonon E' mode. For off-resonance excitations, only the E' peak is strong. However, when the excitation energy is close to excitonic resonances, the 2LA signal overwhelms the E' phonon peak (see Figure 2 below). We believe that a similar mechanism is responsible for the appearance of new peaks in 1~3L samples in the current work. Figure 3(a), which is reproduced from Supplementary Fig. 9, shows that the peak at 210 cm⁻¹ for 1L and 2L is weak for excitation energies of 2.54 and 2.81 eV but becomes very strong for the excitation energy of 2.41 eV. The calculated phonon dispersion in Figure 3(b), which is from Ref. [25], shows that there are several phonon branches near ~105 cm⁻¹ at the zone boundaries. All these observations point to the conclusion that the appearance of the new peaks can be ascribed to resonance-enhanced multi-phonon scattering as in the cases of MoS₂ or WS₂. We have added a sentence to the main text to make this comparison clear.

Figure 1. (a) Excitation energy dependent Raman spectra of monolayer MoS₂ [36, Lee *et al.*, *Nanoscale* **7**, 3229 (2015)]. (b) Calculated phonon dispersion of monolayer and bulk MoS₂ [Molina-Sánchez *et al.*, *Phys. Rev. B* **84**, 155413 (2011)].

Figure 2. (a) Excitation energy dependent Raman spectra of monolayer WS₂ [37, del Corro *et al.*, *Nano Lett.* **16**, 2363 (2016)]. (b) Calculated phonon dispersion of monolayer and bulk WS₂ [Molina-Sánchez *et al.*, *Phys. Rev. B* **84**, 155413 (2011)].

Figure 3. (a) Excitation energy dependent Raman spectra of mono- and bi-layer NiPS₃ [Supplementary Fig. 9]. (b) Calculated phonon dispersion of bulk NiPS₃ [25, Bernasconi *et al.*, *Phys. Rev. B* **38**(17), 12809 (1988)].

Relevant Revisions

- **Changes in manuscript (Page 12, line 20)**

We note that there are several peaks that originate from multi-phonon scattering at ~ 210 , ~ 590 , and ~ 800 cm^{-1} (see Supplementary Fig. 8). Some of these peaks are strongly enhanced due to the resonance effect^{34,35}. For example, the relative intensities or the line shapes of P_3 at ~ 210 cm^{-1} or the peak at ~ 590 cm^{-1} vary with the excitation laser energy, which is a clear indication of resonant processes (see Supplementary Note 2 and Supplementary Fig. 9).

→ We note that **in the spectra for thinner samples** there are several peaks that originate from multi-phonon scattering at ~ 210 , ~ 590 , and ~ 800 cm^{-1} (see Supplementary Fig. 8). Some of these peaks are strongly enhanced due to the resonance effect. **Similar enhancement effects on multi-phonon peaks due to resonance have been observed in other 2D materials**³⁶⁻³⁸. For example, the relative intensities or the line shapes of P_3 at ~ 210 cm^{-1} or the peak at ~ 590 cm^{-1} vary with the excitation laser energy, which is a clear indication of resonant processes (see Supplementary Note 2 and Supplementary Fig. 9).

On the other hand, we agree that the interpretation of the quasi-elastic scattering (QES) signal is not fully satisfactory. As far as we understand, the theory for QES is not well-established yet [23]. Our analysis follows that of Reiter and Halley [41, 42] as was done in Ref. [23]. Although there are still some disagreements, this theory has been successful in explaining experimental observations in 2D $\text{SrCu}_2(\text{BO}_3)_2$ [44] and 1D KCuF_3 [43]. We have added a statement in the main text to make this point clear.

Relevant Revisions

- **Changes in manuscript (Page 15, line 8)**

For a more quantitative analysis, the temperature dependence of the phonon population should be considered. The measured intensity of Stokes-shifted QES is simply expressed by³⁷⁻⁴⁰

→ For a more quantitative analysis, the temperature dependence of the phonon population should be considered. Although the exact mechanism of QES in low-dimensional systems is not fully understood yet²³, we follow the theory of Reiter⁴¹ and Halley⁴² to analyze our data. The measured intensity of Stokes-shifted QES is simply expressed by⁴⁰⁻⁴⁵

Reviewer #2

Comments to the authors

I have read the revised version of the paper by Cheong et al.

I am overall happy with the proposed revisions. I find the new data and the presentation strongly improved and find the present version of the manuscript very convincing.

I do recommend its publication in Nature Communications.

The only minor comment I would have left concerns the terminology 'P2' and 'splitting' for this mode. I find this a bit misleading.

There are in fact two modes, one with Ag and one with Bg symmetry that happen to have the same energy in the non-magnetic state, and a different one in the magnetic state.

I find the discussion very convincing, but the wording 'splitting' is more reminiscent of the degeneracy lifting of a mode, and this is not the case here. So I'd rather refer to a 'magnetic-order-induced energy difference' or to some extra hardening or so than to a 'splitting'.

Authors' Response

We thank the Reviewer #2 for his/her positive comments and an important suggestion. We changed the expression of 'P₂ splitting' to 'phonon frequency difference ΔP_2 ' or 'magnetic-order-induced frequency difference of P₂' in the text and figures.

Relevant Revisions

- Revised Fig. 4, 5 and Supplementary Fig. 12

Reviewer #3

Comments to the authors

The authors have made a valiant effort to clarify the issues that have been raised in the review. The paper now provides a fairly clear view of the capabilities and limitations of Raman scattering for the investigation of antiferromagnetic 2D materials. Nonetheless, publication in Nature Communications is not justified in my opinion, for the following reasons.

First, the Raman spectrum of the one-monolayer sample looks very different from those of thicker samples. The primary phonons are significantly broader, and there are several extra modes. The authors assert that some of the extra modes may be multiphonon excitations, and some indications of extra phonon modes are already visible in two- and three-monolayer samples. On balance, however, the evidence points to a substantial modification of the lattice structure and/or structural domain distribution in the thinnest samples. Especially in view of the strong magneto-elastic coupling, it is implausible to assume that this structural modification and the apparent modification of the magnetic properties are unrelated, and to attribute the degradation of the Neel temperature entirely to spin fluctuations. The central message of this paper is therefore dubious.

Authors' Response

We thank the reviewer for acknowledging our hard work to improve the data. As we explained in our response to reviewer #1, the appearance of additional broad and strong peaks can be reasonably explained in terms of resonance-enhanced multiphonon scattering. The reviewer also pointed out somewhat broader primary peaks. For example, the strongest peak at 385 cm^{-1} indeed appears broader in monolayer. In 2D layered materials, the intra-plane structure is known to be quite robust in the monolayer limit. This has been demonstrated in high-resolution TEM and electron diffraction studies on monolayer graphene and monolayer MoS_2 . To the best of our knowledge, there is no reported case of monolayer having a different intra-plane structure in any 2D layered material. Furthermore, the peak positions of the main Raman peaks are not much different in the monolayer case, indicating that there is no major structural changes going from several layers to monolayer in our samples.

Therefore, we conclude that the intra-plane structure of our monolayer NiPS₃ is not different from that of bulk or few-layer samples.

On the other hand, thinner materials such as one- or two-layer thick samples are more susceptible to the inhomogeneous broadening effects due to the substrate or the exposed top surface. For example, the spatially inhomogeneous distribution of charged centers in the substrate can cause broadening of phonon lines, and thinner samples should be more susceptible to such broadening. Indeed, the main Raman lines in MoS₂ or FePS₃ also show some broadening as the thickness is decreased (see Figure 4 below). However, such broadening rarely affects the principal physical properties of a material much. For example, in the case of Ising-antiferromagnet FePS₃, which has very similar chemical properties and was prepared in much the same way as our NiPS₃ samples, , the antiferromagnetic phase transition temperature was observed at nearly the same temperature although the phonon lines are somewhat broader in monolayer samples, demonstrating that the magnetic ordering is sustained down to the monolayer limit. If the disorder effect was significant, the experimental results should have been different. In the same light, we believe that the main physics of our current work, 'suppression of antiferromagnetic ordering in the monolayer limit' is not a result of external disorders but a fundamental effect of dimensionality.

Figure 4. Normalized Raman spectra of monolayer and few-layer. (a) MoS₂, (b) FePS₃, and (c) NiPS₃.

Comments to the authors

Second, the current study does not open up significant new perspectives for the investigation of antiferromagnetic 2D materials. Two-magnon Raman scattering probes high-energy magnetic fluctuations that are only indirectly related to the antiferromagnetic order parameter. Direct probes of antiferromagnetic order are not in sight, as the authors themselves have pointed out. It is not apparent how studies of this kind can uncover new information on XY or XXZ systems that is not already

known from previous work on this subject. The authors' justification for publication in Nature Communications in the abstract ("This work provides the first experimental examination of the XY magnetism of two-dimensional systems.") is inappropriate as there has been extensive prior work on a variety of platforms ranging from ultrathin magnetic films to Josephson junction arrays. For a summary, see for instance Taroni et al., J. Phys.: Condens. Matter 20, 275233 (2008). Currently there is no discussion at all of prior research on XY systems. A discussion of the context should be added, wherever the paper may eventually be published.

Authors' Response

As the reviewer mentioned, much research has indeed been done on the XY magnetic model as summarized in Taroni *et al.* We thank the reviewer for pointing out this important reference. As Taroni *et al.* summarized, previous work on this topic was done mostly on layered magnetic materials or ultra-thin metal films. Using these systems one can indeed study the XY physics as they exhibit quasi two-dimensionality. However, our work is fundamentally different in that we probe the XY physics in the atomically thin monolayer limit, which is the ultimate 2D limit. This is very much different from studying the two-dimensionality in layered crystals. For example, although NiPS₃ is a layered antiferromagnet in the bulk case, we demonstrate that its behavior is fundamentally different in the monolayer limit. Importantly, this drastic thickness dependence is only shown for XY system of NiPS₃, not the Ising system of FePS₃. This demonstrates that one cannot reveal all the main features of 2D magnetic systems from the study of quasi-2D layered magnetic crystals. In this sense, our work opens a new door to the study of XY magnetism in 2D systems which is truly novel. We have now added Taroni *et al.* as Ref. [8] and provided a brief description of key previous work in this regard.

Moreover, we respectfully disagree with the reviewer's opinion that the lack of direct probe of antiferromagnetic ordering in atomically thin materials diminishes the importance or impact of this work. Our work demonstrates that even with an indirect measurement tool such as Raman spectroscopy, important new physics can be revealed. Lack of a direct probe should not stop us from exploring the 2D magnetism in the atomically thin limit. Until a direct probe is found and fully developed to be used reliably, one should strive to utilize all the available tools to unearth the beautiful physics in this new class of materials.

Relevant Revisions

- **Changes in manuscript (Page 2, line 1)**

This work provides the first experimental examination of the XY magnetism of two dimensional systems and opens new opportunities of exploiting these fundamental, theorems of magnetism using magnetic van der Waals materials.

→ This work provides the first experimental examination of the XY magnetism **in the atomically thin limit** and opens new opportunities of exploiting these fundamental, theorems of magnetism using magnetic van der Waals materials.

- **Changes in manuscript (Page 3, line 11)**

The generic form of the magnetic Hamiltonian can be written as follows⁷:

$$H = - \sum_{\langle i,j \rangle} (J_x S_i^x S_j^x + J_y S_i^y S_j^y + J_z S_i^z S_j^z), \quad (1)$$

where $J_{x,y,z}$ is the nearest-neighbor exchange interaction for spin components, and i and j run through all lattice sites and nearest neighbors, respectively. S_i^x , S_i^y , and S_i^z are the x , y , and z component of the total spin at i -site. For the 2D Ising system

($J_x = J_y = 0$) the magnetic ground state is stable even in the 2D limit, whose experimental evidence has been recently presented using magnetic van der Waals materials: FePS₃^{8,9} with antiferromagnetic order and Cr₂Ge₂Te₆¹⁰ and CrI₃¹¹ with ferromagnetic order.

→ The generic form of the magnetic Hamiltonian can be written as follows⁷:

$$H = - \sum_{\langle i,j \rangle} (J_x S_i^x S_j^x + J_y S_i^y S_j^y + J_z S_i^z S_j^z), \quad (2)$$

where $J_{x,y,z}$ is the nearest-neighbor exchange interaction for spin components, and i and j run through all lattice sites and nearest neighbors, respectively. S_i^x , S_i^y , and S_i^z are the x , y , and z component of the total spin at i -site. The critical behaviors of two-dimensional magnetic systems have been studied using layered magnetic crystals or ultra-thin metal films⁸. However, it is still desirable for one to investigate the major features of 2D magnetism using true 2D materials. For the 2D Ising system ($J_x = J_y = 0$) the magnetic ground state is stable even in the 2D limit, whose experimental evidence has been recently presented using magnetic van der Waals materials: FePS₃^{9,10} with antiferromagnetic order and Cr₂Ge₂Te₆¹¹ and CrI₃¹² with ferromagnetic order.